# Analyzing glacier retreat and mass balances using aerial and UAV photogrammetry in the Ötztal Alps, Austria

Joschka Geissler[1,4], Christoph Mayer[2], Juilson Jubanski[1], Ulrich Münzer[3] & Florian Siegert[1]

[1]3D Reality Maps GmbH, Dingolfinger Str. 9, Munich, 81673, Germany

[2]Bavarian Academy of Science, Geodesy and Glaciology, Alfons-Goppel Str. 11, Munich, 80539, Germany

[3]Ludwig-Maximilians-University, Department of Earth and Environmental Sciences, Section Geology (remote sensing), Luisenstr. 37, 80333 Munich, Germany

[4]Faculty of Environment and Natural Sciences, Albert-Ludwigs University Freiburg, Friedrichstr.39, 79098 Freiburg, Germany

*Correspondence to*: Florian Siegert (Siegert@realitymaps.de)

**Abstract.** We use high-resolution aerial photogrammetry to investigate glacier retreat in great spatial and temporal detail in the Ötztal Alps, a heavily glacierized area in Austria. Long-term in-situ glaciological observations are available for this region as well as a multitemporal time series of digital aerial images with a spatial resolution of 0.2 m acquired over a period of 9 years. Digital surface models (DSMs) are generated for

the years 2009, 2015, and 2018. Using these, glacier retreat, extent, and surface elevation changes of all 23 glaciers in the region, including the Vernagtferner, are analyzed. Due to different acquisition dates of the large-scale photogrammetric surveys and the glaciological data, a correction is successfully applied using a designated unmanned aerial vehicle (UAV) survey across a major part of the Vernagtferner. The correction allows a comparison of the mass balances from geodetic and glaciological techniques – both quantitatively and spatially.

The results show a clear increase in glacier mass loss for all glaciers in the region, including the Vernagtferner, over the last decade. Local deviations and processes, such as the influence of debris cover, crevasses, and ice dynamics on the mass balance of the Vernagtferner are quantified. Since those local processes are not captured with the glaciological method, they underline the benefits of complementary geodetic surveying. The availability of high-resolution multi-temporal digital aerial imagery for most of the glaciers in the Alps provides

opportunities for a more comprehensive and detailed analysis of climate change induced glacier retreat and mass loss.

# 1    Introduction

The impacts of climate change are widespread and clearly visible in the Alps (Rogora et al., 2018) but particularly evident in the dwindling glacier resources (Beniston et al., 2018; Sommer et al., 2020; Zekollari et al., 2019). Over the past 100 years, the temperature in the European Alps, hereafter referred to as the Alps, has increased almost twice as fast compared to the global average, resulting in nearly 2 °C higher mean air temperatures (Auer et al., 2007; Marty and Meister, 2012). By the end of this century, mean air temperatures are expected to rise further by several degrees Celsius (Gobiet et al., 2014; Hanzer et al., 2018). Due to this ongoing climate evolution, alpine glaciers may lose half of their volume by 2050 compared to 2017 (Zekollari et al., 2019). The response of glaciers to climatic variations is related to the glacier mass balance that can be measured, among others, directly, using the glaciological method, or indirectly, using the geodetic method. For the latter, the volume change of a glacier is determined by integrating the elevation change between two surveys across the entire glacier surface. Volume change is then converted to mass change by incorporating a density assumption of the affected volume. For retrieving geodetic mass balance data, different remote sensing methods exist, varying in the platform (e.g. satellite, airplane, UAV) and sensor (e.g. Laser Scanner, Optic Camera, Radar) used. Their specific benefits and limitations are analyzed and discussed in different studies (Baltsavias et al., 2001; Bamber and Rivera, 2007; Kääb, 2005; Kargel et al., 2013; Pellikka and Rees, 2010).

Glaciological mass balances date back far into the last century and our knowledge of long-term glacier evolution is based on these data sets (Mayer et al., 2013a; WGMS, 2020). However, glaciological mass balances are limited to a few glaciers only, due to the large effort involved in the field work. Existing historic aerial imagery can also provide valuable information on long term glacier evolution and, depending on the imagery, allow a retrospective determination of geodetic glacier mass balances for a considerable number of glaciers and thus greatly complement glaciological data (Belart et al., 2019; Jaenicke et al., 2006; Magnússon et al., 2016; Mayer et al., 2017). Different studies demonstrate the potential of spatiotemporal change analysis of alpine glaciers using photogrammetric data (Fugazza et al., 2018; Gudmundsson and Bauder, 1999; Legat et al., 2016; Rossini et al., 2018) and comparing their results to glaciological mass balances (Baltsavias et al., 2001; Klug et al., 2018). Comparing geodetic and glaciological mass balances generally has the potential to reveal systematic errors and regions of anomalous mass balance conditions (Fischer, 2011). However, when comparing these methods one needs to account for different error sources such as differences in survey dates, errors related to the density assumption, ice dynamics, internal and basal melt and other systematic or random error sources (Pellikka and Rees, 2010; Zemp et al., 2013).

Zemp et al. (2013) provides a framework to perform homogenization, error assessment and calibration of the geodetic and glaciological data sets that is often used in the scientific community (Andreassen et al., 2016; Klug et al., 2018). Building on this framework, considering the required additional data is available, we see further potential (i) regarding the involved density assumption as well as (ii) for extrapolating geodetic mass balances to full mass balance years. These small improvements increase the detail of our geodetic mass balance and thus allow a more in-depth comparison of the geodetic and glaciological mass balances and will be presented within this study.

Regarding volume to mass conversion (i), the simple density assumption provided by Huss (2013) is frequently used to derive overall geodetic mass balances (Andreassen et al., 2016; Belart et al., 2019). However, this assumption is only valid for periods longer than three to five years, medium to high volume change and stable

mass balance gradients (Huss, 2013, Zemp et al, 2013). These strong limitations are often overcome by using firn densification models (Reeh, 2008), existing field datasets (Huss et al., 2009) or pixel-based classifications of snow, firn and ice and assigning a density to each class (Pelto et al., 2019). However, those methods require extensive field measurements and computational effort. We present an easy applicable and transferrable approach, incorporating the equilibrium line altitude (ELA) of a glacier to generate an altitude-related density assumption with a linear transition around the ELA from firn to ice density. The required ELA can be retrieved from satellite imagery (Rabatel et al., 2005) or historic photographs (Vargo et al., 2017).

For the comparison of geodetic and glaciological data, extrapolating the mass balances to full mass balance years (30[th] September) (ii) is required, since survey dates usually differ. Such corrections can for instance be applied by using a simple degree-day model (Belart et al., 2019) or field measurements (Fischer et al., 2011). These methods, however, are either not suitable for retrospective corrections where no field data was collected or do not account for the spatially distributed, glacier specific accumulation and ablation patterns of each glacier (Huss et al., 2009). We present a workflow for using an UAV survey with 5 cm spatial resolution, in combination with a simple degree-day model to correct differences in survey dates between geodetic and glaciological data. This robust approach allows a more detailed analysis of the ice dynamics and the remaining systematic differences between the geodetic and glaciological mass balances.

This study is focused on a study site within the Ötztal Alps, Austria. Airborne photogrammetric datasets covering 23 glaciers are provided by the Austrian Bundesamt für Eich- und Vermessungswesen (BEV) and 3D RealityMaps GmbH for 2009, 2015 and 2018, thus covering a period of 9 years. One of the surveyed glaciers is the Vernagtferner, a reference glacier in the World Glacier Monitoring Service (WGMS) system. Glaciological mass-balances have been determined here using the glaciological method since 1965, while a series of historical maps back to 1889 demonstrates the long-term glacier evolution over more than a century (Escher-Vetter et al., 2009).

Within this study, we describe the full photogrammetric workflow applied to our aerial imagery. After the coregistration of the resulting DEMs, volume changes and general glacier retreat is analyzed for all 23 glaciers. A more detailed analysis is conducted for the Vernagtferner, where our altitude-related density assumption and the UAV-based correction of survey dates allows a spatial and quantitative comparison of the geodetic with the glaciological mass balances.

## 2    Study Area

The Ötztal Alps are located in the central-eastern Alps and represent one of Austria's most extensive glacierized regions, covering a range in altitude between 1700 to 3768 meters above sea level (m.a.s.l.) and more than 250 km² (Fig. 1). It combines the upper regions of the drainage basins of Rofental, Pitztal, and Kaunertal. The location in the inner part of the Alps leads to relatively low precipitation amounts (Fliri, 1975, 186-197), which for example, reach mean values of 660 mm yr[-1] at the valley station Vent in 1969-2006 (Abermann et al., 2009).

For some of the 23 glaciers within the study site, glaciological mass balance measurements exist. One of the longest series of measurements can be found at the Vernagtferner, where regular monitoring by the Bavarian Academy of Sciences and Humanities (BAdW) began in 1965 (Escher-Vetter et al., 2009). This glacier is characterized by several sub-basins, which were connected to a single glacier tongue in former times. In 2018, the glacier covered almost 7 km² within an altitude range between 2860 m.a.s.l. and 3570 m.a.s.l.. The mass

balance is determined by the glaciological method, using measurements at ablation stakes, manual and geophysical depth soundings, and retrieving information from snow and firn pits. The annual and winter mass balances are determined independently of each other by measurements on the fixed dates of 1st May and 30th September, the dates of the glaciological balance year (Cogley, 2010; Cogley et al., 2011). Besides, there is a

long history of geodetic mapping at the Vernagtferner dating back to 1889 (Mayer et al., 2013a).

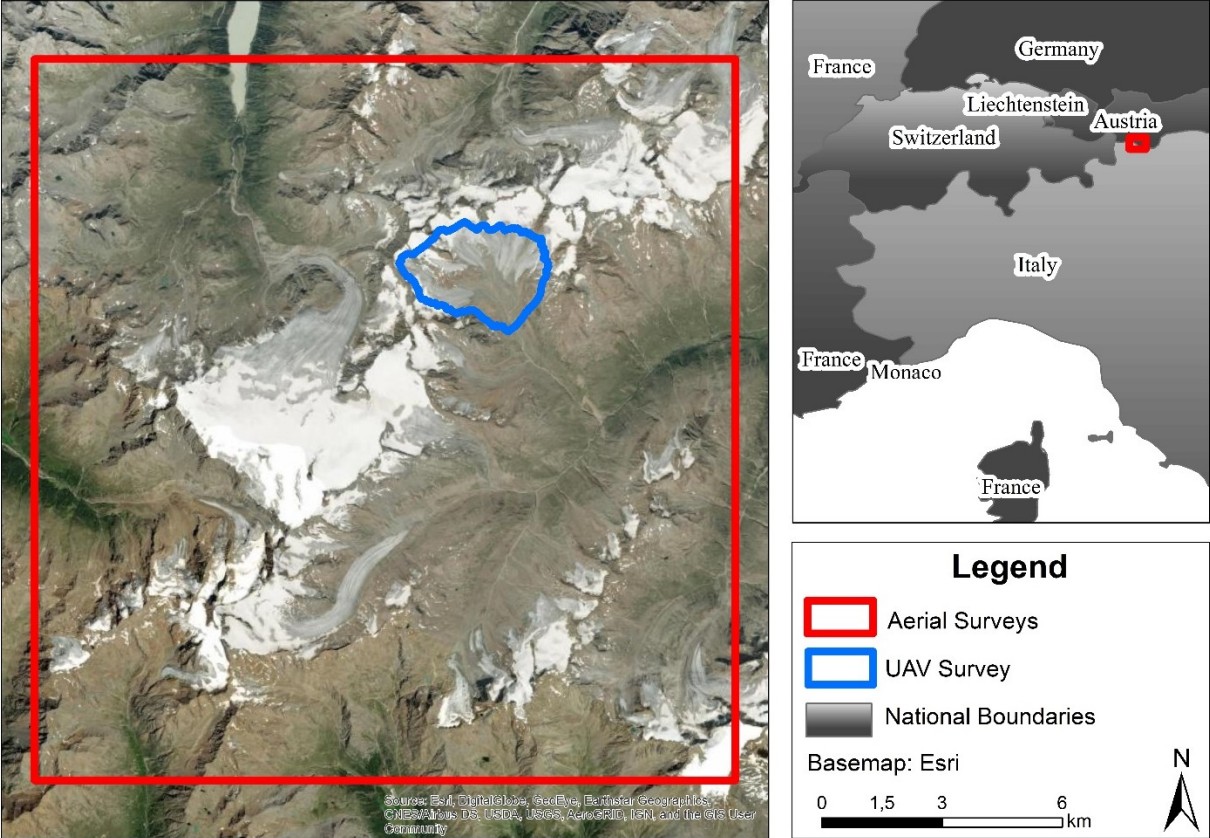

**Figure 1: Study site in the Ötztal Alps; Red: area covered by aerial surveys; Blue: area of the Vernagtferner covered by the UAV Survey; Image source: ESRI (2020)**

### 3    Data Acquisition

### 3.1    Photogrammetric data

In Austria, cadastral aerial surveys are conducted by the BEV with a nominal resolution of 0.2 m. We use a BEV survey from 2015 as a basis for our investigations. Besides, surveys performed by 3D RealityMaps in 2009 and 2018 with the same or higher ground resolution are investigated. Table 1 shows the most relevant information on the conducted air surveys.

In addition to the airplane-based surveys, a smaller test site was covered by an UAV flight to retrieve high-resolution data at another acquisition date closer to the maximum of ablation in 2018 (Table 1). The processed area covers 6 km² containing most of the Vernagtferner, including all glacier tongues. The glacier was almost snow-free at the time of the acquisition, which provides optimal processing conditions.

**Table 1: Overview of the aerial data acquisitions**

| Date | Platform | Area [km²] | Overlap (forward:side) [%] | Resolution [cm] | Images [Count] | Image type | Camera |
|------|----------|-----------|---------------------------|-----------------|----------------|------------|--------|
| 09.09.2009 | Airplane | 257 | 80:40 | 20 | 381 | TIF RGB 8bit | UltraCam XP |
| 03.08.2015 | Airplane | 330 | 80:50 | 20 | 572 | TIF RGBI 16bit | UltraCam XP |
| 21.09.2018 | Airplane | 260 | 80:60 | 20 | 428 | TIF RGBI 16bit | UltraCam Eagle Mark 2 |
| 21.08.2018 | UAV | 6 | 80:80 | 5 | 1992 | JPEG RGB 8bit | UMC-R10C |


## 3.2    Glaciological mass balance data

Glaciological mass balance data is gathered as stake readings from ablation stakes for estimating the ice melt across the glacier. Snow depth and last season firn deposits are determined from mechanical depth soundings with metal probes, which are then combined with density information from snow and firn pits to calculate the

water equivalent of the remaining snow and firn cover. While stake readings only require two length measurements per stake with an uncertainty of typically about 1 cm, more significant errors are included in the direct accumulation measurements due to uncertainties in the sample volume (about 5%) and the determination of density by using spring scales (about 4%). Therefore water equivalent accuracy is about 6% (Zemp et al., 2013). The typical number of stakes used for the annual mass balance measurements at the Vernagtferner is

about 35, while 4-5 accumulation measurements are collected at the end of the glaciological year, 30[th] September.

Glacier boundaries for delineating the spatial mass balance distribution are derived from aerial surveys repeated roughly every decade, updated in the ablation region by annual GNSS (Global Navigation System Services) measurements of the glacier tongue geometry. The spatial error of these measurements is usually better than 1 m.

The information from the stake readings, the depth soundings, the snow and firn pits, and the location of the equilibrium line are combined to interpolate the spatial distribution of the glacier mass balance into a raster file. Due to the sparse information in the accumulation region, it is necessary to manually correct the interpolation results in this region with the knowledge of the long-term accumulation patterns, which are rather persistent. Errors introduced by uncertainties in the accumulation region, however, are relatively small, especially during

the recent decade where the accumulation area ratio (AAR) is usually well below 30 %. While ablation varies between 0 and up to 4.5 m w.e. $a^{-1}$ in the ablation area, accumulation only varies between 0 and about 0.3-0.4 m w.e. $a^{-1}$ in the accumulation area. Within this study, we assumed the error of the interpolated glaciological raster to be 0.1 m. w.e. $a^{-1}$, which is in accordance with Zemp et al. (2013). It must be noted that this relatively large error within the accumulation area will only affect the final mass balance by less than 2 %.

The ELA is derived by comparing oblique terrestrial photographs of the transient snow line and firn extent with optical remote sensing information close to the field measurements date. The derived ELA has a horizontal location accuracy of about 10 m.

## 4    Methods

### 4.1    Photogrammetric workflow

To determine geodetic glacier mass balances from aerial and UAV images, we used a workflow consisting of two main modules: the data processing (Sect. 4.1.1) and the vertical change analysis (Sect. 4.1.2; Fig. 3). The main goal of the data processing module was the reconstruction from raw imagery data into 3D point clouds. Digital Surface Models (DSM) and orthophotos were then derived from these point clouds. Finally, DSM differences were computed. Within the vertical change analysis module, geodetic glacier mass balances were
computed from the DSM differences.

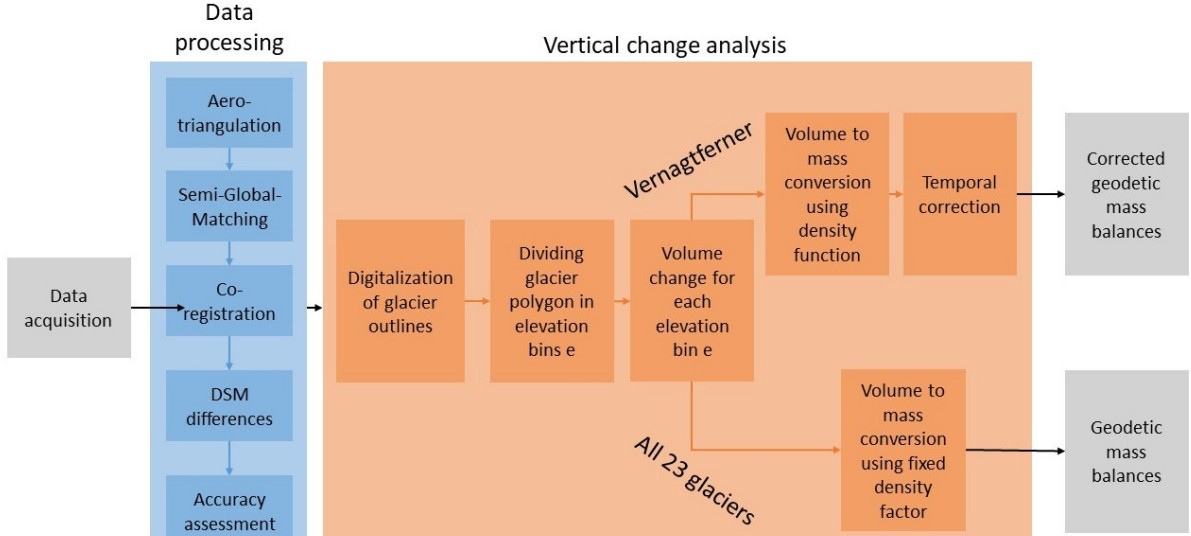

**Figure 2: Full workflow; Blue: From raw images to DSM differences; Orange: From DSM differences to overall and altitudinal mass balances**

### 4.1.1    Data processing

The first step in data processing was the aerotriangulation, which consists of the orientation of the aerial imagery to the real terrain. Modern photogrammetric survey systems deliver highly accurate positions using GNSS and orientation (IMU, inertial measurement unit) information for each image, which were also included in the aerotriangulation for the georeferencing. Finally, at least 20 tie points were manually identified for each image
block to enhance the orientation accuracy. For the large format imagery, the aerotriangulation was performed using the software *Match-AT* by Trimble. For the UAV imagery, the software *Metashape Professional* from Agisoft was used.

The second stage of data processing was the generation of point clouds through three-dimensional reconstruction. For this purpose, the state-of-the-art Semi-global-matching algorithm (Heipke, 2017;
Hirschmüller, 2019) was used. This algorithm was implemented within the software *SURE* (nFrames), which generates DSM and orthophotos from the point clouds. The horizontal shift of the derived orthophotos and

DSMs was computed based on ground control points from the BEV and lies between 10 and 20 cm depending on the acquisition year and thus within the ground resolution of the images. Due to these excellent values, only a vertical coregistration of the DSM differences was applied. Therefore, existing systematic height shifts between all DSMs were derived using 50 stable points (e.g., solid rock) outside the glaciers. 21 stable points were used for the UAV-survey. The 2015 DSM was chosen as the reference because it was derived from the official Austrian cadastral survey and is referenced to the Austrian national survey system. Based on this mean vertical shift over stable ground, all DSMs except for the reference DSM were adjusted in height relative to the reference DSM of 2015. Subsequently, the DSM differences 9/2018–8/2015, 8/2015–9/2009, 9/2018–9/2009, and 9/2018–8/2018 were computed.

### 4.1.2 Vertical change analysis

The goal of the vertical change analysis (Fig. 2, orange part) was to quantify elevation changes $\Delta h_t$ within our study area from the DSM differences of different time periods t. By integrating $\Delta h_t$ over a specific area $S_t$, volume change $\Delta V$ was determined by the following equation, where r is the pixel size (Fischer, 2011; Zemp et al., 2013):

$$\Delta V_t = r^2 * \int_0^{S_t} \Delta h_t \tag{1}$$

To derive overall geodetic glacier mass balances $B_{geod,t}$, $\Delta V_t$ was determined with $S_t$ being the area at the beginning of the respective period t (Fischer, 2011). $S_t$ was digitized visually using orthophotos at a scale of 1:2000. For the following volume to mass conversion (Zemp et al., 2013), we used the density assumption proposed by Huss (2013) ($\bar{\rho}$= 850 kg m$^{-3}$ ± 60 kg m$^{-3}$) for all glaciers.

$$B_{geod,t} = \frac{\Delta V}{\bar{S}} * \frac{\bar{\rho}}{\rho_{water}} \quad \text{with} \quad \bar{S} = \frac{S_{t=begin}+S_{t=end}}{2} \tag{2}$$

To allow a detailed analysis of the altitudinal dependencies of the glacier mass balances, the glacier area was divided into 10 m elevation bins e. For each bin, the geodetic mass balance $B_{geod,e,t}$ was derived with the same equations (see Eq. 1, 2 and Zemp et al. (2013)).

In contrast to most of the glaciers within our study area, the ELA (Sect. 3.2.) of the Vernagtferner is known and lies at 3217 m.a.s.l. for the period 2009-2015, 3278 m.a.s.l. for the period 2015–2018 and, averaged over the entire study period, at 3237 m.a.s.l. for 2009-2018. Thus, we were able to use an altitude-related density function $\bar{\rho} = f_{t,d}(e)$ for converting surface changes to mass relative to the altitude of the ELA for the Vernagtferner (Figure 2). This density function $f_{t,d}(e)$ [kg m$^{-3}$] represents the gradual change from ice density (900 kg m$^{-3}$) in the ablation region to firn density (550 kg m$^{-3}$ (Cogley et al., 2011)) in the accumulation region with elevation e, by using a linear transition zone of ±50 m around the ELA of the respective period t:

$$f_{t,d}(e) = \begin{cases} 550 & \text{for e} > \text{ELA} + 50 \text{ m} \\ 725 - 3.5 * (e - ELA_t) & \text{for e between ELA} \pm 50 \text{ m} \\ 900 & \text{for e} < \text{ELA} - 50 \text{ m} \end{cases} \tag{3}$$

### 4.2 Comparison with glaciological data

To allow a comparison of the geodetic and glaciological mass balances, both datasets were reanalyzed independently according to the steps 1 to 4 in Zemp et al. (2013): Datasets were homogenized (Sect. 3.2 and 4.1.1.), annual glaciological mass balances were accumulated to the periods 09/2009-09/2015, 09/2015-09/2018,

and 09/2009-09/2018, mean annual mass balances of the respective periods (Sect. 3.2. and 4.1.2) as well as systematic and random errors for all geodetic datasets (Sect. 4.3) were derived (Nuth and Kääb, 2011). Because one main objective of this paper was to analyze systematic differences between the two methods, iterative adjustment and calibration of the data (step 5-6, Zemp et al. (2013)) was not performed.

We developed a workflow to account for the remaining temporal differences between the photogrammetric and glaciological mass balance data. Correction periods were defined between the survey date of the photogrammetric data and the end of the glaciological year (Table 2). An additional photogrammetric DSM difference was derived for one month within the ablation period in 2018 (t=corr) using an UAV survey (Table 1). Therefrom, a regression (sigmoid, see Fig. 3) was performed for deriving the altitude dependent surface elevation change $\Delta h_{k,t=corr}$ (Table 2) of the respective period t=corr (Table 2). By multiplying $\Delta h_{t=corr}$ with the altitude related density assumption $f_{t,d}(e)$ (Eq. 3) the mass balance $B_{geod,e,t=corr}$ for all elevation bins e and the correction period t=corr (Table 2) was derived. Only those altitude levels were considered for the regression, which were at least 40 % covered by the UAV survey. The standard deviation (SD) of the regression is 0.07 m ice (Fig. 3). Above 3180 m.a.s.l., thus, for the accumulation areas, the correction function is not based on any data and is therefore error-prone.

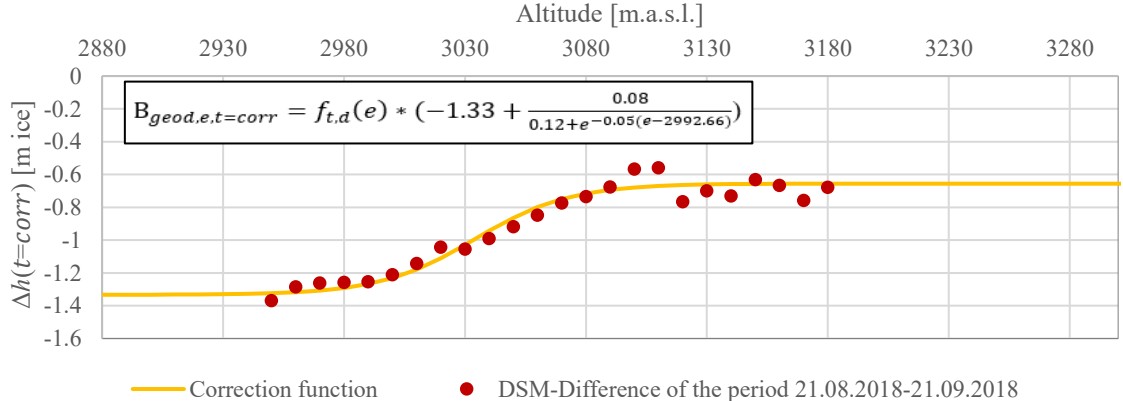

**Figure 3: Surface changes in 10 m altitude bins for the correction period (21.08.2018 to 21.09.2018, red dots); Regression curve (yellow), representing height changes related to the altitude; SD of the regression is 0.07 m ice.**

To transfer this information to the correction periods (t=1-3, Table 2), we used temperature time series measured at a climate station close to the Vernagtferner at 2640 m.a.s.l.. Positive Degree Day Sums ($PDD_{e,t}$) [°C] were computed for all elevation bins e and time periods t (Table 2). To determine the Degree Day Function (DDF) for all elevation bins, we assumed the vertical lapse rate of the air temperature to be -0.6 °C per 100 meters of altitude (Eq. 4). More information on the DDF method can be found, for instance, in Braithwaite and Zhang (2000) and Hock (2003).

$$DDF_e = \frac{B_{geod,e,t=corr}}{PDD_{e,t=corr} * d_{t=corr}} \tag{4}$$

The geodetic mass balance for the correction periods $B_{e,t}$ could then be determined for the time periods t of length $d_t$ and in all elevation bins e:

$$B_{geod,e,t} = DDF_e * PDD_{e,t} * d_t \tag{5}$$

Subsequently, the geodetic mass balances of the full study periods $B_{geod,e, 09/2009 – 08/2015}$, $B_{geod,e, 08/2015 – 09/2018}$, and $B_{geod,e, 09/2009 – 09/2018}$ were corrected according to Table 2 and recalculated to an annual basis. In this study, we

refer to these temporally corrected geodetic mass balances as annual geodetic mass balances (unit: m. w.e. a$^{-1}$) and provide the information on the period with full years (e.g. 2009-2018). Uncorrected geodetic mass balances are referred to by year and their respective month (e.g. 09/2009-09/2018).

**Table 2 Correction parameters (left) of all correction periods t and the applied corrections to all geodetic mass balances (right).**

| t | Correction periods | | $d_t$ [days] | $PDD_{e=2870\ m.a.s.l.,\ t}$ [°C] | Applied Corrections |
|---|---|---|---|---|---|
| 1 | 09.09.2009 | 30.09.2009 | 21 | 82.45 | $B_{geod,e,09-15} = B_{geod,e,\ 09/2009-08/2015} - B_{geod,e,t=1} + B_{geod,e,t=2}$ |
| 2 | 03.08.2015 | 30.09.2015 | 59 | 280.76 | $B_{geod,e,15-18} = B_{geod,e,\ 08/2015-09/2018} - B_{geod,e,t=2} + B_{geod,e,t=3}$ |
| 3 | 21.09.2018 | 30.09.2018 | 9 | 46.58 | $B_{geod,e,09-18} = B_{geod,e,\ 09/2009-09/2018} - B_{geod,e,t=1} + B_{geod,e,t=3}$ |
| corr | 21.08.2018 | 21.09.2018 | 31 | 176.81 | |

The overall geodetic mass balances $B_{geod,\ t}$ of the Vernagtferner was then derived from the mass balances of the single elevation bins e and their respective area $S_{e,t}$ (Zemp et al., 2013):

$$B_{geod,t} = \frac{\sum_{e=1}^{E} B_{geod,e,t} * S_{e,t}}{S_t} \qquad (6)$$

Finally, accumulated glaciologically derived rasters $B_{glac,t}$ (Sect. 3.2) were subtracted from the adjusted geodetic mass balances $B_{geod,t}$. The resulting Variation Rasters $Var_t$ show the spatial deviations between the two methods, where negative values occur for areas where $B_{glac,t} > B_{geod,t}$ and positive values for the opposite relation.

$$Var_t = B_{geod,t} - B_{glac,t} \qquad (7)$$

Using this Variation Raster, we analyzed spatial deviations between both methods that occur due to differences in the individual methods and related errors (e.g., neither including the supra-glacial debris cover or crevassed areas for surface ablation nor dynamic processes within the glaciological method). Therefore, we digitized the respective areas on the Vernagtferner by using the geodetically derived orthophotos of 2009 and 2018 and computed the mean annual variation (difference between the annual geodetic and the glaciological mass balances) by using the Variation Raster (Eq. 7) as well as Eq. 1 and 2. By comparing this mean variation with the mean variation of areas within the same elevation bin, we estimated the magnitude of error introduced by neglecting those areas within glaciological mass balances.

### 4.3    Vertical accuracy assessment

To assess the error distribution within the study area after the coregistration, temporally uncorrected geodetic DSM differences were analyzed following the methodology presented by Nuth and Kääb (2011). Therefore, 1.5 km² of ice-free, stable terrain, representing a wide range of topography was digitized manually. The mean shift and the SD within those areas were calculated, and their relation to topography was investigated. To estimate the error when averaging over extended areas, we followed Rolstad et al. (2009) by assessing the spatial covariance of the elevation differences using semivariograms. Thus, we derived range-values from the semivariograms of all periods and converted those to the confidence intervall of the respective DSM difference. For more detailed information on this method, see Rolstad et al. (2009). For the application of the method, we assumed that

elevation differences are constant in space, do not contain any large-scale trends, and that there is no significant variation of the variance in space.

Basic error propagation was implemented (Nuth and Kääb, 2011; Zemp et al., 2013) to determine the compound error of DSM differences, density conversion (7 %, Sect. 4.1.2, Huss (2013)), the correction function of the acquisition dates (SD = 0.07 m ice a$^{-1}$, Sect. 4.2) as well as the error associated to the glaciological interpolation raster (Sect. 3.2) for all presented results. The error within this study is indicated by the 95% confidence interval.

## 5    Results

### 5.1    Vertical Change Analysis

#### 5.1.1    Visual assessment

Using the derived glacier outlines, orthophotos, and DSM differences, the first results are obtained by a visual interpretation for the entire study area. In general, glaciers have thinned and reduced in size. For instance, surface height changed at the glacier tongue of the Hintereisferner by up to – 20.4 ± 0.4 m during the nine years from 09/2009 to 09/2018 (Fig. 4). With increasing altitude, the loss of height on the glacier approaches zero. Surface elevation changes around the glacier tongue of the Hintereisferner can be attributed, among others, to local debris movements and an existing dead-ice body (Fig. 4). Analyzing the orthophotos of the three surveys reveals that the eastern part of the Hochjochferner lost a considerable area along its lower glacier margin. Especially its main tongue shortened by 826.4 ± 0.2 m between 2009 and 2018 (Fig. 5, white dotted circle). Further visualization of the DSM differences can be assessed at https://og.realitymaps.de/AlpSense/.

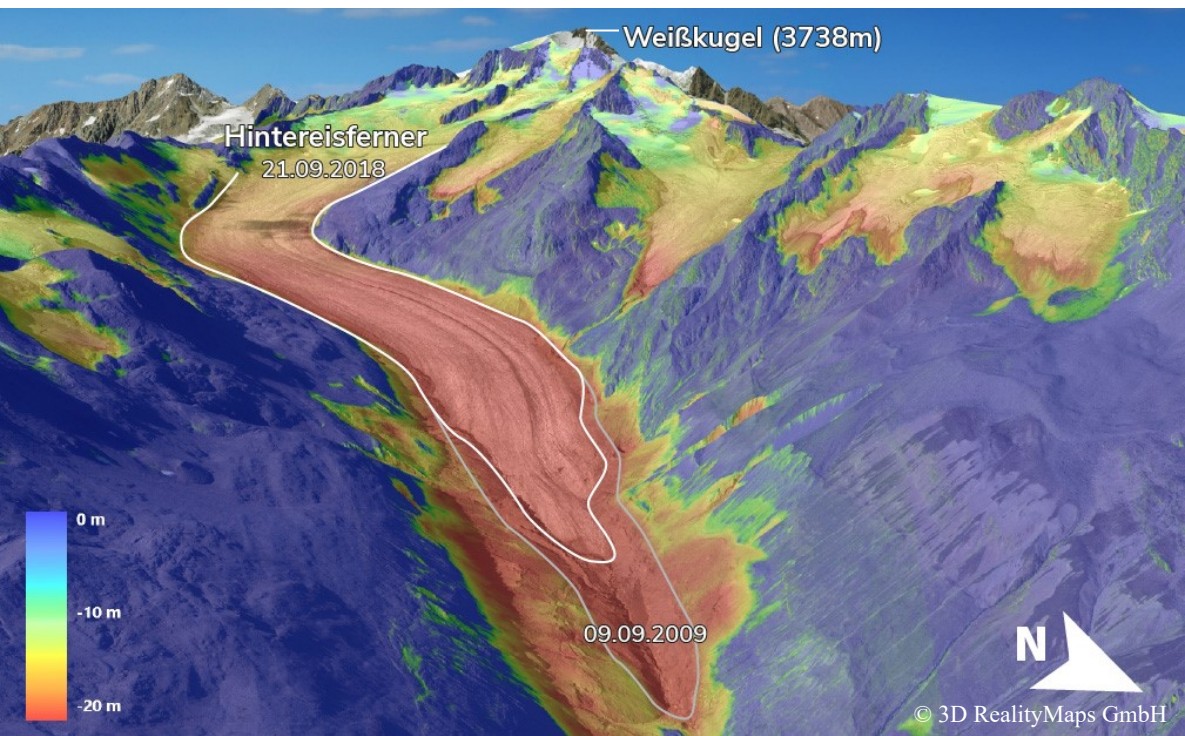

**Figure 4: 3D representation of the color-coded height differences in meter (m) between the DSMs of 09/2009 and 09/2018 for Hintereisferner (white outline). Surface elevation loss appears in red, constant elevation appears blue. Ice loss is especially high at altitudes below 2500 m. The underlying orthophoto was derived from the 2018 aerial survey. Height loss in the south-east of the glacier tongue can likely be attributed to an existing dead-ice body.**

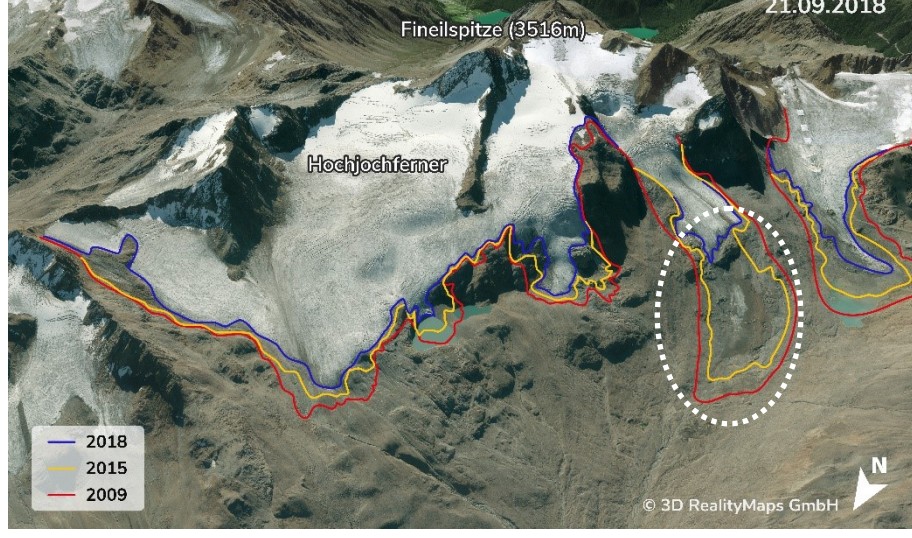

**Figure 5: 3D visualization of the length change for the Hochjochferner in 2009 (top), 2015 (middle), and 2018 (bottom); Lines indicate glacier extent for the respective years. The second glacier tongue from the right (white dotted circle) lost 826 m in 9 years.**

### 5.1.2 Vertical accuracy assessment

The accuracy assessment (Sect. 4.3) allows an estimation of potential error related to the DSM differences and derived products. In general, the mean vertical error of the DSM differences ranges from -0.16 m to 0.10 m, SD does not exceed 0.42 m (Fig. 6a) and errors are generally smaller for the period 08/2015-09/2018. As expected, SD increases with the slope for both periods (Fig. 6b, e). An apparent increase of the SD can also be found in lower altitudes (< 2600 m.a.s.l., (Fig. 6d, g), most likely due to the increasing influence of vegetation. A relation between aspect and the vertical error was found for the DSM difference 09/2009 – 08/2015 (Fig. 6c), which can be attributed to a horizontal shift of the DSM 2009 (Sect. 4.1.1) (Nuth and Kääb, 2011).

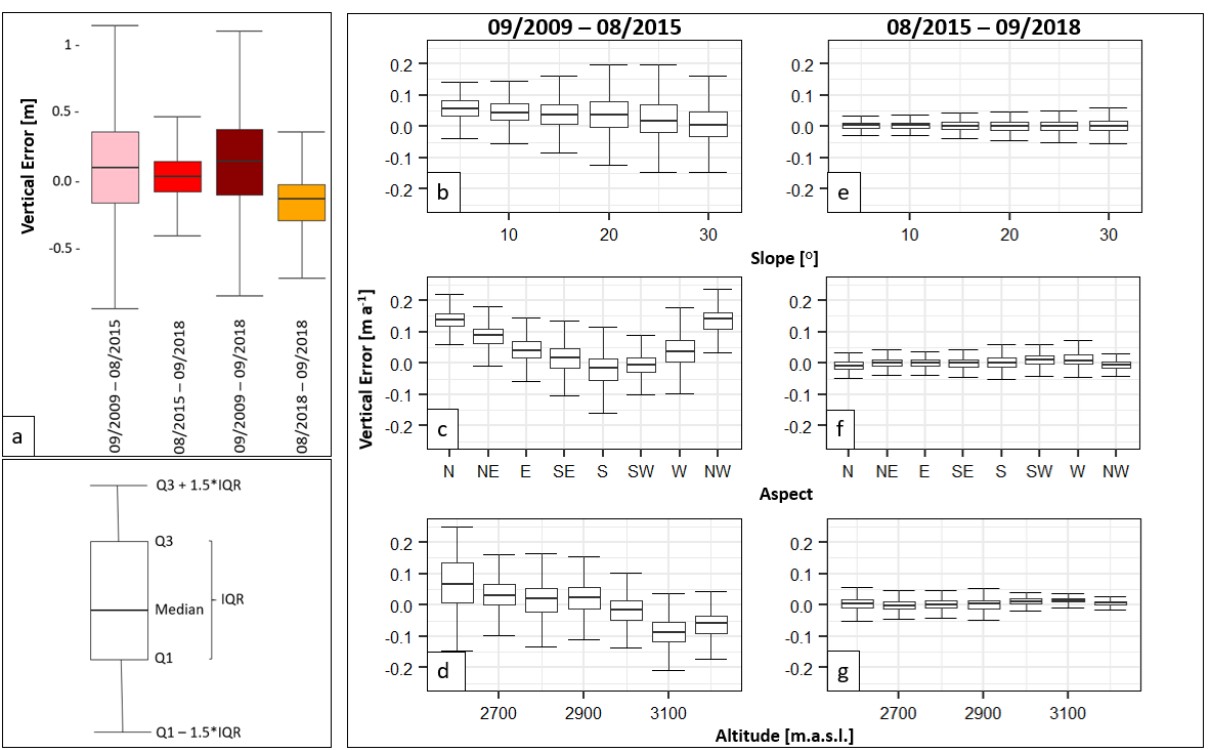

**Figure 6: Overall vertical error of the DSM differences (a); Annual vertical error for the periods 09/2009-08/2015 (b-d) and 08/2015-09/2018 (e-g) in relation to slope (b,e), aspect (c,f) and altitude (d,g).**

### 5.1.3 Mass balances

Volumetric changes $\Delta V_{t,e}$ and height changes $\Delta h(t,e)$ for every elevation bin e were derived for all periods t and all glaciers, before and (for the Vernagtferner only) after the correction of acquisition dates (Eq. 1 – 2, Fig. 7).

Before and after the temporal correction, the annual height changes per elevation bin show a characteristic shape with height changes being the most negative at the glacier tongues. The period between 2009 and 2015 generally shows less negative values than the period 2015 and 2018 in all altitudes for all glaciers (Fig. 7, bottom left) and the Vernagtferner (Fig. 7, top left). Although after the temporal correction, the difference in height change between the two periods is considerably smaller.

Compared to the height changes, the most negative volume change occurs in higher elevations since it is linked to the area-height-distribution of a glacier. After correcting for the acquisition dates, based on a regression curve (Gaussian fit) for the values of the Vernagtferner, between 2009 and 2015, the most negative volume change of -0.21 ± 0.01 million m³ a⁻¹ occurred at an altitude of 3009-3019 m.a.s.l.. Between 2015 and 2018, the most

negative volume change further decreased to -0.29 ± 0.02 million m³ a⁻¹ at an increased altitude of 3089-3099 m.a.s.l. (Fig. 7, top right). The same trend was found for all glaciers within the study area, although no correction was applied to those values (Fig. 7, bottom right).

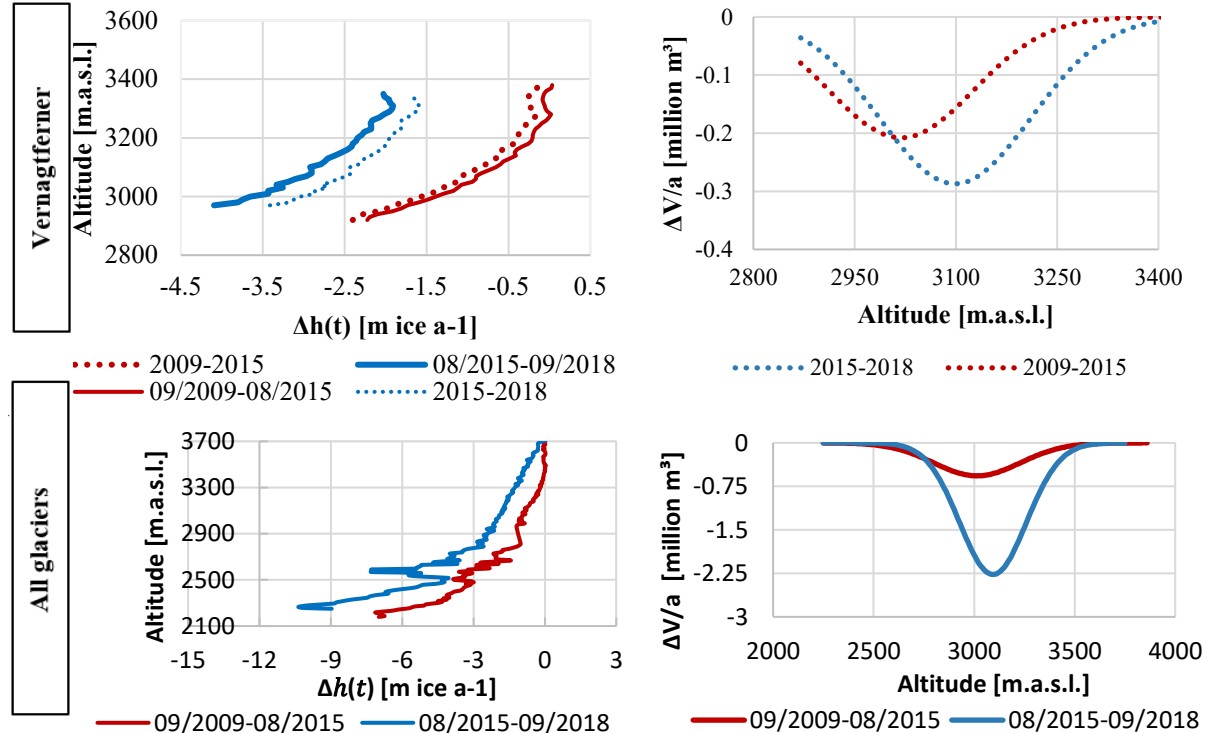

Figure 7: Top left: Annual height changes Δh(t) per altitude for the Vernagtferner, Bottom left: Surface height changes of all glaciers; Right: annual volume changes for the Vernagtferner (top) and all glaciers (bottom); Dotted lines represent the surface or volume changes after the correction of the acquisition dates.

## 5.2  Comparison with glaciological measurements

The geodetic and glaciological mass balances of the Vernagtferner were further compared to detect discrepancies in spatial distribution or magnitude between both methods. As already noted, the overall mass balance for the Vernagtferner is more negative in the period 2015-2018 than in the period 2009-2015. This can be seen in both, the glaciological and the geodetic mass balances (Fig. 8). The uncorrected geodetic mass balance (blue) does not equal the corresponding glaciological mass balance (orange). After the correction of acquisition dates (red), the annual geodetic mass balance has approached the glaciological data. However, the corrected data is still more negative for 2015-2018 and 2009–2018. For the period 2009-2015, corrected geodetic and glaciological mass balances match.

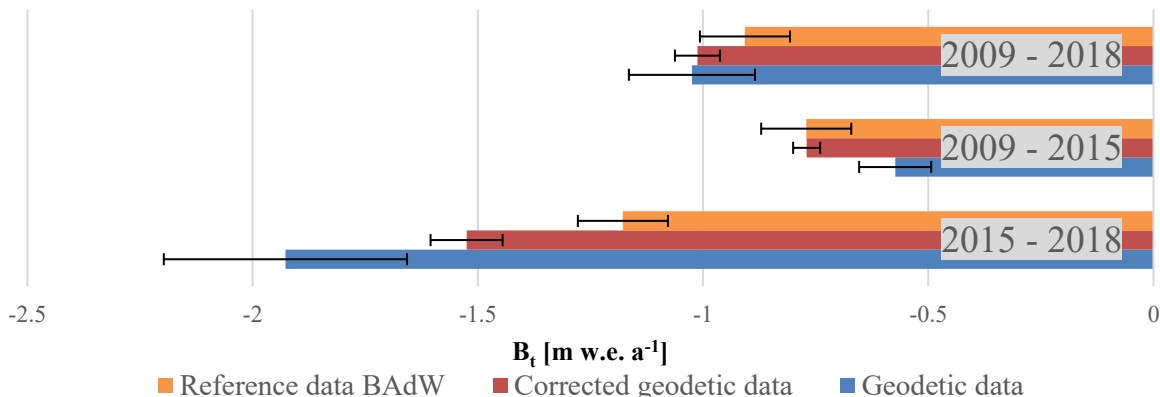

**Figure 8: Comparison between geodetic mass balances (blue) and glaciological mass balances (orange) for the investigation periods. Red bars visualize corrected geodetic data.**

The spatial differences between both methods were further investigated using the Variation Raster described in sect. 4.2.. It is noticeable that the Variation Raster is more positive (thus $B_{glac,t} < B_{geod,t}$, see Eq. 7) in debris-covered areas compared to its surrounding cells (red in Fig. 10). Since no ablation stakes are located in supra-glacial debris at the Vernagtferner and glaciological mass balances are interpolated from surrounding information on clean ice, the effect of debris is neglected within the glaciological method. Debris cover above a certain thickness protects the glacier against incoming heat fluxes (Östrem, 1959) and therefore reduces the ablation. With our Variation Raster, we were able to quantify the effect of neglecting debris-covered areas within the glaciological interpolation: For 2018, a total area of 91350 m² (1.5 % of the total area of the Vernagtferner) at a mean altitude of 3040 m.a.s.l. was classified as debris cover. The variation between the geodetic and glaciological method for those debris-covered areas is $0.56 \pm 0.3$ m w.e. a$^{-1}$ and thus more positive than the mean variation at the respective altitude $0.38 \pm 0.27$ m w.e. a$^{-1}$ (period 2009–2018, see Fig. 9). Considering the difference of those values, the overall glaciological mass balance of 2009-2018 would be 0.17 m w.e. a$^{-1}$ (2 %) less negative if debris-covered areas were considered in the interpolation.

Further analysis of the Variation Raster (Fig. 10) emphasizes that geodetic and glaciological mass balances do not only differ due to debris cover or other local effects. More precisely, it is evident that the deviations depend on the altitude. Thus, in the lower parts of the glacier, the geodetic mass balance is less negative than the glaciological mass balance. For higher altitudes, the geodetic mass balance is more negative than the glaciologically derived one. This pattern is usually attributed to the ice dynamic component of elevation change contained in the geodetic differences (submergence and emergence of ice and firn). Additionally, Fig. 9 clearly shows that comparing the different observation periods, the bias between both methods becomes larger within the accumulation areas.

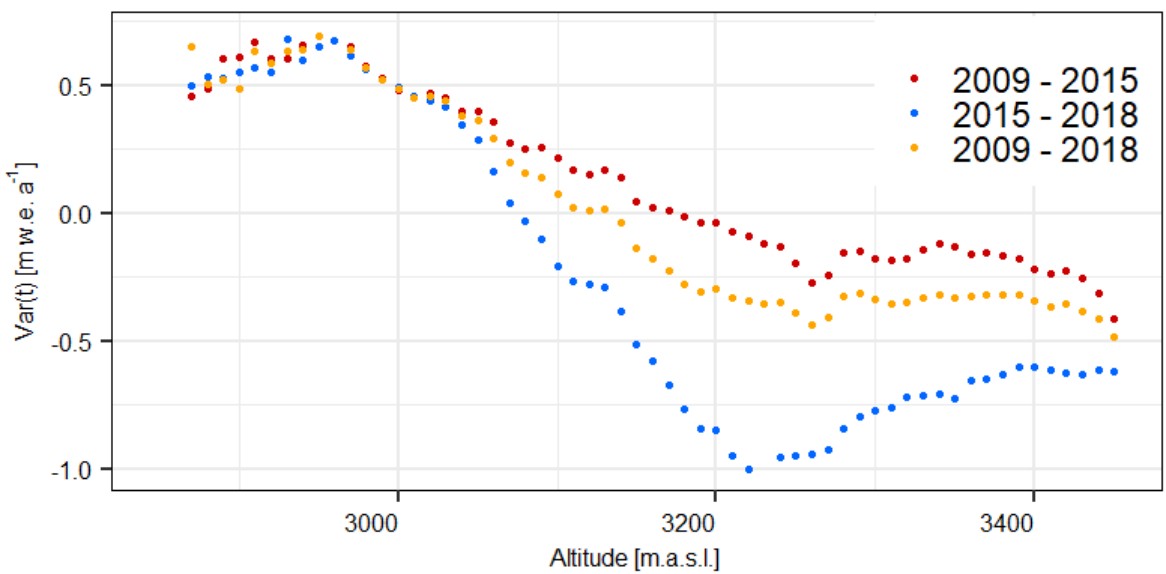

**Figure 9: Methodical variations between annual geodetic (corrected) and glaciological mass balances, depending on the altitude [m w.e. a$^{-1}$] for all periods t; Positive values are present where $B_{glac,t} < B_{geod,t}$, negative values are present for the opposite relation, see Eq. 7.**

Assuming all variations between geodetic and glaciological derived mass balances (Fig. 9) are caused by

dynamic processes, the magnitude of emergence and submergence is computed using the Variation Raster and Eq. 1 and 2. Between 2009 and 2018, the mean emergence (between 2900 m.a.s.l. and 3050 m.a.s.l.) is 0.55 ± 0.25 m w.e. a$^{-1}$, with a most positive value being 0.7 ± 0.25 m w.e. a$^{-1}$. Submergence occurs at altitudes higher than 3150 m.a.s.l. and results in a mean of - 0.31 ± 0.25 m w.e. a$^{-1}$, with the most negative value being -0.48 ± 0.25 m w.e. a$^{-1}$ (Figure 10 and Table 3). For a glacier in balance, the change between submergence and

emergence regions occurs close to the equilibrium line. However, the mean ELA of the Vernagtferner lies at 3237 m.a.s.l. for the period 2009-2018 and thus roughly 150 m higher than the switch between apparent emergence and submergence.

When comparing the five different accumulation basins of the Vernagtferner the derived submergence is more negative for higher basins with the largest offsets for the remaining true accumulation regions Hochvernagt and

Taschachhochjoch (Figure 10 and Table 3).

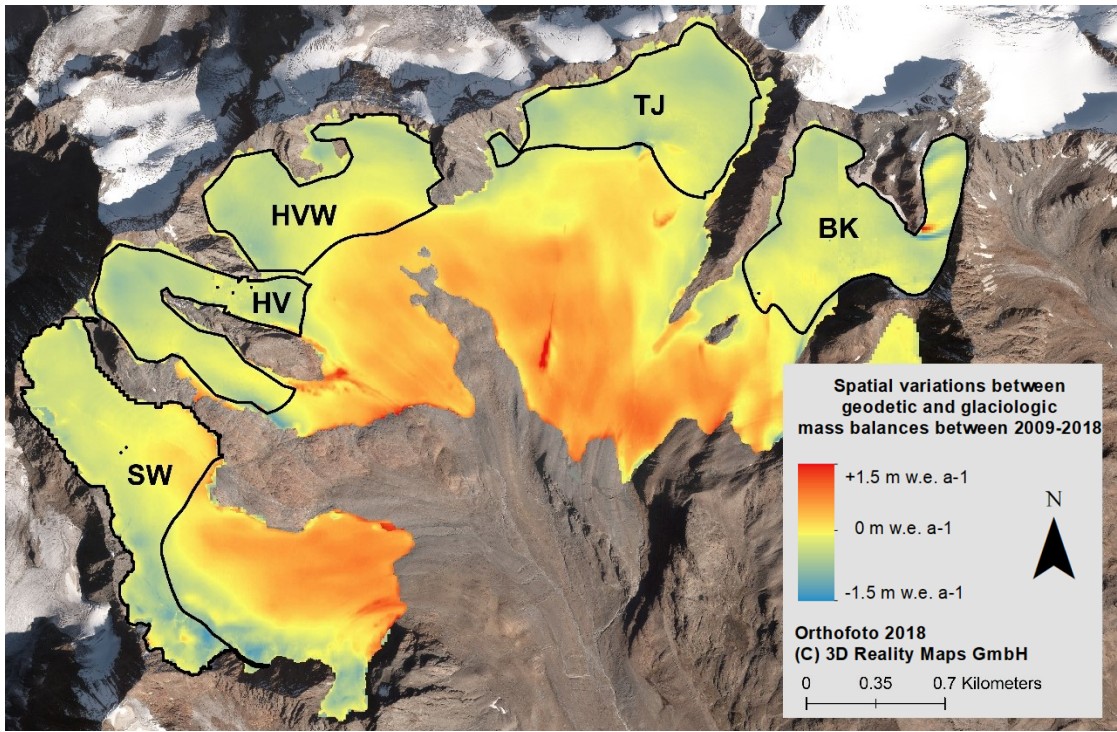

Figure 10: Top: Variation Raster 2009–2018: Spatial variations of the annual geodetic (corrected) and glaciological mass balance data; Blue areas with negative variations represent areas where $B_{geod,t} < B_{glac,t}$. In red areas, the opposite relation is present (see Eq. 7). For regions that appear yellow, both methods present similar mass balances; Black outlines represent the different accumulation areas of the Vernagtferner.

Table 3: Mean values of the Variation Raster 2009-2018 (Fig. 10) for the accumulation areas in m w.e. a$^{-1}$, interpreted as submergence; Associated error is ± 0.25 m w.e a$^{-1}$ for all submergence values.

| Accumulation Area | | Mean submergence between 2009 and 2018 [m w.e. a$^{-1}$] | Area > 3150 m.a.s.l. [km²] |
|---|---|---|---|
| SW | Schwarzwand | - 0.26 | 0.69 |
| HV | Hochvernagt | - 0.38 | 0.44 |
| HVW | Hochvernagtwand | - 0.30 | 0.54 |
| TJ | Taschachhochjoch | - 0.41 | 0.56 |
| BK | Brochkogel | - 0.28 | 0.62 |
| **VN** | **Vernagtferner (total)** | **- 0.32** | **2.85** |

## 5.3 Other glaciers within the study site

The major advantage of geodetic mass balances compared to glaciological mass balances is that large areas can be investigated in great spatial detail. To highlight this benefit, we investigated the geodetic mass balance of all glaciers within the study area (Sect. 4.1.2). The temporally uncorrected geodetic mass balance of all glaciers is -0.46 ± 0.06 m w.e. a$^{-1}$ between 09/2009 and 08/2015, while it quadruples to -1.59 ± 0.21 m w.e. a$^{-1}$ between 08/2015 and 09/2018. For the aggregated period (09/2009–09/2018), the uncorrected geodetic mass balance of all glaciers is -0.84 ± 0.11 m w.e. a$^{-1}$. It must be noted that the quadrupling of the geodetic mass balance is partly due to the temporal correction that was not carried out. Thus, the mass balances do not refer to the glaciological year and cannot be compared with glaciological data. If a time correction is assumed to have a similar influence on the mass balances of other glaciers as it had for the Vernagtferner (-34 % for 09/2009-08/2015, +20 % for

08/2015–09/2018, and +1.2 % for 09/2009–09/2018, Fig. 8), the annual mass balances of all glaciers can be

estimated. Accordingly, the annual glacier mass balance of all glaciers within the study area would be -0.61 ±

0.07 m w.e. a⁻¹ (2009-2015), -1.25 ± 0.14 m w.e. a⁻¹ (2015-2018) and -0.83 ± 0.1 m w.e. a⁻¹ (2009-2018). Based

on this assumption, annual geodetic mass balances would have doubled between the two periods 2009-2015 and

2015-2018 and the annual geodetic mass balance of the Vernagtferner for the same periods is 25 %, 21 %, and

22 % more negative than the mean of all glaciers (Fig. 8).

The remaining error due to the horizontal shift of the 2009 DOM (Sect. 5.1.2) is negligible for these general

considerations but must be taken into account when working with individual mass balances (Table 4, Fig. 11).

**Table 4: Surface changes [m] and geodetic mass balances [m w.e. a⁻¹] of all glaciers within the study area. Geodetic**
**mass balances were derived by using a fixed density factor (Sect. 4.1.2. and Figure 2); Values are temporally**
**uncorrected and thus do not match the glaciological year; Glaciers marked with (*) must be considered with**
**additional caution for the periods 09/-2009-08/2015 and 09/2009-09/2018 due to their orientation. Related errors are**
**higher than the error estimates presented in this manuscript (Sect. 5.1.2.)**

| Name | $\Delta h_k$ 09/2009 - 08/2015 [m] | $\Delta h_k$ 08/2015 - 09/2018 [m] | $\Delta h_k$ 09/2009 - 09/2018 [m] | $B_{geod}$ 09/2009 - 08/2015 [m w.e. a⁻¹] | $B_{geod}$ 08/2015 - 09/2018 [m w.e. a⁻¹] | $B_{geod}$ 09/2009 - 09/2018 [m w.e. a⁻¹] | Area 2009 [km²] | Area 2015 [km²] | Area 2018 [km²] |
|---|---|---|---|---|---|---|---|---|---|
| Eisferner | -3.29 | -6.20 | -9.48 | -0.47 | -1.76 | -0.90 | 0.84 | 0.77 | 0.63 |
| Gepatschferner | -2.72 | -5.15 | -7.87 | -0.39 | -1.46 | -0.74 | 18.95 | 18.78 | 18.59 |
| Guslarferner mi. | -4.54 | -6.71 | -11.24 | -0.64 | -1.90 | -1.06 | 1.56 | 1.42 | 1.27 |
| Hintereisferner | -5.18 | -6.85 | -12.03 | -0.73 | -1.94 | -1.14 | 7.13 | 6.44 | 6.10 |
| Hintereiswände | -2.98 | -4.88 | -7.86 | -0.42 | -1.38 | -0.74 | 0.41 | 0.41 | 0.34 |
| H. Ölgrubenferner | -1.00 | -4.23 | -5.23 | -0.14 | -1.20 | -0.49 | 0.06 | 0.06 | 0.05 |
| Hochjochferner | -5.26 | -5.81 | -11.08 | -0.75 | -1.65 | -1.05 | 4.90 | 4.34 | 3.84 |
| Kesselwandferner | -1.44 | -4.81 | -6.25 | -0.20 | -1.36 | -0.59 | 3.70 | 3.60 | 3.56 |
| Kreuzferner M  (*) | -2.97 | -5.05 | -8.02 | -0.42 | -1.43 | -0.76 | 0.34 | 0.33 | 0.18 |
| Kreuzferner N1 (*) | -2.25 | -5.23 | -7.48 | -0.32 | -1.48 | -0.71 | 0.27 | 0.25 | 0.21 |
| Kreuzferner S (*) | -3.53 | -4.97 | -8.50 | -0.50 | -1.41 | -0.80 | 0.56 | 0.53 | 0.46 |
| Langtaufererferner | -3.84 | -6.52 | -10.36 | -0.54 | -1.85 | -0.98 | 3.07 | 3.04 | 2.89 |
| Mitterkar Ferner | -2.61 | -3.93 | -6.54 | -0.37 | -1.11 | -0.62 | 0.54 | 0.54 | 0.52 |
| Ö. Wannetferner | -3.25 | -6.33 | -9.58 | -0.46 | -1.79 | -0.91 | 0.58 | 0.58 | 0.42 |
| Rofenberg E (*) | -3.15 | -4.46 | -7.62 | -0.45 | -1.26 | -0.72 | 0.09 | 0.09 | 0.07 |
| Rotkarferner (*) | -2.55 | -4.20 | -6.75 | -0.36 | -1.19 | -0.64 | 0.22 | 0.22 | 0.11 |
| Sayferner (*) | -1.95 | -4.91 | -6.86 | -0.28 | -1.39 | -0.65 | 0.27 | 0.27 | 0.25 |
| Sexegertenferner | -1.96 | -5.43 | -7.39 | -0.28 | -1.54 | -0.70 | 2.98 | 2.91 | 2.46 |
| Taschachferner | -1.14 | -4.24 | -5.38 | -0.16 | -1.20 | -0.51 | 5.82 | 5.78 | 5.30 |
| Taschachferner E. | -3.57 | -4.15 | -7.72 | -0.51 | -1.18 | -0.73 | 0.05 | 0.05 | 0.05 |
| Vernaglwandferner | -3.72 | -5.94 | -9.66 | -0.53 | -1.68 | -0.91 | 0.75 | 0.70 | 0.58 |
| Vernagtferner | -4.05 | -6.80 | -10.85 | -0.57 | -1.93 | -1.03 | 7.33 | 7.02 | 6.10 |
| W.H. Ölgrubenferner | -2.07 | -4.72 | -6.79 | -0.29 | -1.34 | -0.64 | 0.14 | 0.14 | 0.13 |

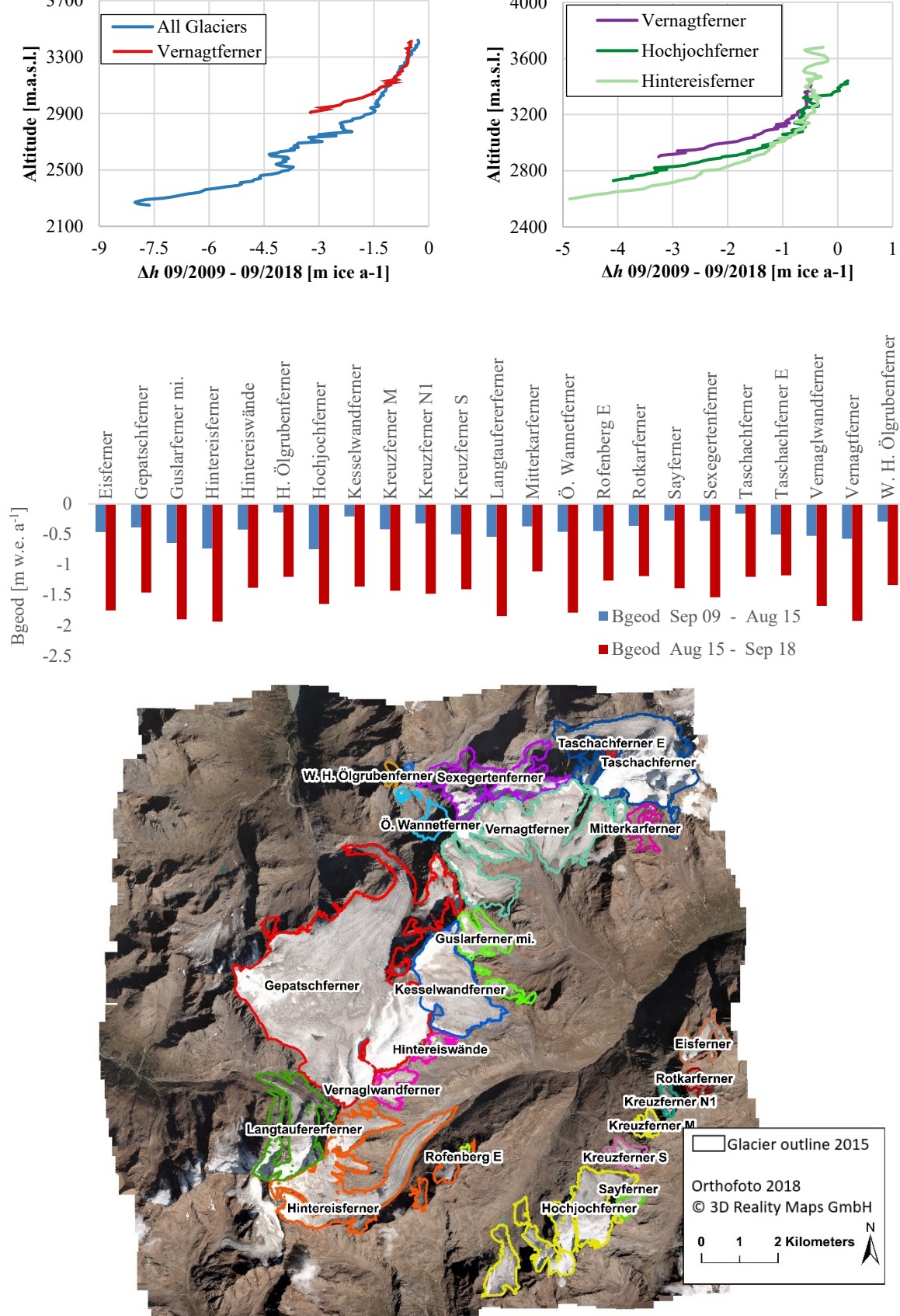

**Figure 11: Annual height changes per altitude between 09/2009 and 09/2018 for the Vernagtferner compared to (i) all glaciers within the study area (top left) and (ii) to the Hochjochferner and Hintereisferner (top right); Comparison of temporally uncorrected geodetic mass balances of all glaciers within the study area for the two investigation periods (middle); glacier outlines of 2015 shown on orthophoto 2018 (bottom).**

## 6    Discussion

This study investigated multitemporal changes in surface heights of selected glaciers in the Ötztal Alps with high spatial resolution. We derived geodetic mass balances for 23 glaciers within our study area and found that glacier mass loss accelerated within the last decade. At the same time, the absolute surface height changes showed a maximum at low altitudes for all glaciers (Fig. 7). Peak volume loss occurs at higher altitudes, compared to the altitude where maximum surface height losses occur since the volume is directly linked to the area-height-distribution of a glacier. These findings correspond to general observations (i) for the Vernagtferner (BAdW, 2019; Escher-Vetter, 2015) and (ii) for most of the glaciers in the Alps (Davaze et al., 2020; Fischer et al., 2015b; Fischer et al., 2015a; Huss, 2012).

We further evaluated the vertical surface height changes as a function of elevation (Fig. 7). Even though they cannot be directly compared to glaciologically derived vertical balance profiles, they illustrate the relation of height losses to altitude (Kaser et al., 2003; Pellikka and Rees, 2010). We found that the elevation changes of all glaciers within our study site (i) are on average less negative compared to the Vernagtferner, (ii) are correlated to the altitude of the glacier tongue, and (iii) increased between 09/2009–08/2015 and 08/2015–09/2018 (Fig. 11). Besides glacier analysis, surface elevation changes caused by other processes (e.g. dead-ice bodies, debris movement) can be identified using high-resolution photogrammetric data. For instance, on the north-facing side of the valley, below the glacier tongue of Hintereisferner, existing surface elevation changes (Fig. 4), that can mostly be attributed to an existing dead ice body, were analyzed: Over an area of $27 * 10^4$ m², volume loss was quantified to be $(1.3 \pm 0.22) * 10^6$ m³ for the period $2009 - 2018$.

The error assessment revealed the general high precision of photogrammetric data with the SD of the vertical error not exceeding 0.42 m for all DSM differences (Fig. 6). The variation of the vertical error is normally distributed and semivariograms determined to assess the error for averaged values show a sill for all periods (further information on the method can be found in Rolstad et al. (2009)). However, a relation was found for the vertical error regarding the aspect for the period 09/2009-08/2015 and 09/2009-09/2018, which is presumably based on a horizontal shift of the DSM 2009 (Sect. 4.1.1). Thus, assumptions made hereupon (Sect. 4.3.) are potentially less valid, meaning that the relation to the aspect can cause larger error bars then we suggest. Especially for glaciers with homogeneous orientation to the north or northwest, presented values must be considered with caution. However, only a few glaciers are affected by this and are highlighted accordingly in this manuscript. By applying a full coregistration, following  Nuth and Kääb (2011), this rotational error could have been addressed.

We used the density assumption of Huss (2013) to convert height changes to water equivalent for all glaciers except for the Vernagtferner, that is repeatedly used for volume to mass conversions in other studies (Andreassen et al., 2016; Belart et al., 2019). All DSM differences fulfill the prerequisites for this density assumption (Huss, 2013). However, the DSM difference 08/2015 – 09/2018 being a rather short period of only three years just meets those requirements. Thus, the density assumption and the derived geodetic mass balances for this period must be considered with caution. For the Vernagtferner, we applied an altitude-related density function based on the known ELA. Since the density of the glacier volume lost is related to the ELA, our approach is more suitable for shorter time periods where the influence of annual meteorological conditions is increased, compared to the static density assumption provided by Huss (2013). For the Vernagtferner, ELA increased 61 m between our two study periods 2009 – 2015 and 2015 – 2018. This relatively small change in elevation changed the AAR of this

glacier by about 15% and thus has an influence on the density of the volume lost during those periods that should not be neglected. The presented density function accounts for such changes of the AAR and thus allows a more detailed spatial comparison of the geodetic and glaciological mass balances. Since the ELA can be estimated from satellite imagery (Rabatel et al., 2005), our method is transferable to other glaciers. We recommend, however, adjusting the firn density values to the corresponding region. The presented altitude-related density function is easily applicable and needs low computational effort and therefore greatly complements other existing density conversion factors, that account for e.g. firn compaction processes, rely on classification methods or modelling (Pelto et al., 2019; Reeh, 2008).

Derived geodetic mass balances (Table 4) are well in accordance with the literature. For instance, the derived geodetic mass balance of the Hintereisferner ($-1.14 \pm 0.02$ m w.e. a$^{-1}$, Table 4) for the period 09/2009 – 09/2018 differs by only 8 % compared to the glaciological mass balance of the same period ($-1.24$ m w.e. a$^{-1}$ (WGMS, 2020)), measured by the Institute of Atmospheric and Cryospheric Sciences of the University in Innsbruck. However, since geodetic surveys are not always possible at the end of the glaciological year, there are large deviations (especially for the shorter periods 09/2009-08/2015 and 08/2015-09/2018) from the glaciological periods and thus a direct comparison with glaciological mass balances is not possible. As a result, the mass balances shown in Table 4 are influenced by the temporal offset, which causes parts of the observed increase in mass balance magnitude. This must be considered for any further use.

We developed a methodology to account for this temporal offset thus to adjust geodetic mass balances to the end of the glaciological year (30$^{th}$ September). An additional UAV survey was conducted, which allowed assessing vertical height changes during one month of the ablation period in 2018 (Fig. 3). Therefrom, a regression function (sigmoid) was determined to estimate surface changes relative to the elevation. We incorporated meteorological data from a nearby climate station by using a simple degree-day model for all relevant periods and elevation bins. Our regression function includes uncertainty for the accumulation areas since the UAV survey did not cover the entire glacier area. However, we showed that the presented workflow provides good results as corrected annual geodetic mass balances agree well with glaciological mass balances (Fig. 8). Those results suggest that our method is suited for retrospective corrections of geodetic mass balances and can improve comparisons of geodetic with glaciological mass balances. However, the method will perform best if the AAR and area-height distribution remain similar between the period to be corrected and the period of the determined correction function. Thus, the retrospective correction is limited to a few years only. The presented method complements other existing correction methods that rely, for instance, on field data, available for only a few glaciers, or DDF-models that do not have spatially explicit output (Belart et al., 2019; Fischer, 2011). It must be noted that our method was only tested on the Vernagtferner for a limited number of observation periods and thus further testing is required to assess its robustness and transferability to other glaciers and periods. Therefore, individual correction functions must be determined for each glacier individually, as these are directly related to glacier-specific slope gradients, orientation, the height of the glacier tongue, and area-height distributions (Fig. 11). To determine such individual correction functions, temperature information (for the correction period as well as all periods to which the correction is applied) and an additional geodetic survey is needed to estimate the surface elevation changes during the correction period. The length of the correction period was one month within this study, however we do not expect poorer results if varying this period by 1-2 weeks. The required geodetic survey is neither limited to a platform, nor to a sensor. Thus, all geodetic survey configurations that allow the

determination of geodetic glacier mass balances from DSM differences are suited for deriving the presented correction function. We used a dedicated photogrammetry-based UAV survey, flexible and low-cost, that enabled a correction with high spatial resolution.

Comparing the corrected geodetic with interpolated glaciological mass balance rasters revealed local deviations (Fig. 10). They mainly occur in crevassed and debris-covered areas and thus agree with findings of other studies (Fischer, 2011; Pellikka and Rees, 2010). They originate in the lack of glaciological ablation stakes that are not located in such areas, and thus the glaciological method interpolates within those regions. For crevassed areas, the deviations found at the Vernagtferner are negligible. Regarding the influence of debris cover for the period

2009 - 2018, we found that glaciological mass balance would be 2 % less negative if such regions would be considered within interpolation. Debris cover might play an increasing role when debris-covered glacier areas increase due to further glacier decline (Scherler et al., 2018). This increases the need to conduct geodetic observations in addition to glaciological measurements because point measurements on debris-covered parts are not always representative.

Surprisingly, the differences between geodetic and glaciological volume change in the accumulation areas do not reveal large spatial deviations, even though the glaciological results rely on very sparse in-situ data. This demonstrates that the long-term accumulation conditions at Vernagtferner, which are known from former detailed investigations (Mayer et al., 2013a; Mayer et al., 2013b), are spatially relatively stable and can be used for scaling the point measurements. However, this comparison of geodetic and glaciological results in the

accumulation region has the potential to improve the representation of spatial variability across these areas.

The deviations between the geodetic and glaciological mass balances also showed a clear dependency on the elevation (Fig. 9) for all periods. These deviations are constant for the ablation areas for all periods but vary for the accumulation areas by a factor of three. We assumed that internal and basal melting is negligible, and the influence of temporal snowpack differences is small due to the long period of investigation and the rather small

ratio of accumulation area. We thus interpret the variations as the large-scale dynamic processes submergence and emergence. They become quantifiable when looking at the mean variations per altitude of both methods (Fig. 9 and 10). It is noticeable that the observed switch between emergence and submergence (Fig. 9) is about 100 m below the ELA of the Vernagtferner for the respective period 2009-2018 (BAdW, 2019) and roughly 200 m below the ELA for the period 2015-2018. While the ELA increased between the two periods (2009-2015 vs.

2015-2018) by 61 m, the change from submergence to emergence decreased by about 130 m. The reasons for this development are unknown but might indicate a further deviation from balanced conditions during recent years. The derived submergence must be considered with high caution since it is neither validated with field measurements nor model results. Presented submergence values are also based on (i) our applied correction with sparse data for the accumulation region, (ii) on the interpolation of a few glaciological stake readings, and (iii)

our assumption that internal and basal processes are negligible. Those values (Table 3) are thus error-prone and should only be referred to as a rough estimation. However, they show differences between the individual accumulation areas, which support glaciological observations (Lambrecht et al., 2011; Mayer et al., 2013b).

Future aerial image acquisitions aiming at calculating glacier mass balances must be as close as possible to each other (same date within the year) and preferably at the end of the ablation season. Ideally, these acquisitions are

taken close to the standard glaciological mass balance data of 30[th] September to allow a direct comparison with available field information. It should be ensured that the survey covers the entire surface of the glacier and that

derived DOMs are fully coregistered. Since aerial image surveys require cloud-free weather, it will not always be possible to acquire the images on this exact date. This paper presented a method to account for the resulting differences in acquisition dates and to project geodetic mass balances to full glaciological periods.

For future research in the presented field, we see great potential in reanalyzing glacier mass balances using existing aerial imagery. Since the late 2000s, European state surveying agencies have been carrying out cadastral surveys every two to four years with digital sensors with forward overlap of 80% and side overlap of 40-60 %, thus creating an immense multitemporal database of aerial imagery suitable for such purposes. The scientific and commercial consideration of those imageries would allow a three-dimensional reconstruction, mapping, and

multitemporal analysis of vast areas in the Alps with a resolution in the order of decimetres.

## 7     Conclusion

This study demonstrates the potential of aerial images and the resulting DSM's for analyzing glacier retreat in great spatial and temporal detail. For all 23 glaciers within the study area, geodetic mass balances were derived with 0.2 m spatial resolution. It was shown that glacier retreat does not only take place in low altitudes, but that

even high accumulation basins are meanwhile affected due to non-compensated ice flow. The elevation of the highest glacier height changes and the amount of maximum volume loss increased within our observation period (2009-2018). In the future, if all aerial surveys that are available for large parts of the Alps were used, the majority of all alpine glaciers could be analyzed with the presented method. The spatial resolution of the respective analysis would be significantly better than in other recent satellite-based studies. We compared the

photogrammetric results with glaciological in-situ measurements of the Vernagtferner. An altitude-related density assumption was used to account for the rise of the ELA and to increase the spatial detail of the comparison with the glaciological data. Geodetic mass balance periods were adjusted to the glaciological survey dates by using an additional UAV survey, combined with a simple degree-day model. This correction as well as our density function allowed a spatially explicit, retrospective determination and correction of geodetic mass

balances and is transferable to other glaciers. The results show that the glaciological method can be greatly complemented with photogrammetric analyses, increasing the accuracy of the glaciological mass balance series, revealing regions of anomalous mass balance conditions, and allowing estimates of the imbalance between mass balance and ice dynamics.

*Data availability.* Except for the aerial imagery and derived products, all datasets are available on request. Geodetic mass balance data will be submitted to WGMS.

*Author contributions.* Conceptualization, Methodology, Investigation, Formal Analysis and Writing – Original Draft: JG; Validation: CM; Photogrammetric data acquisition and processing: JJ and FS; Glaciological data

acquisition and processing: CM; Writing – Review & Editing: JG, CM, UM, JJ and FS; Funding Acquisition: FS and UM; Supervision: CM, UM, JJ, FS. All authors have read and approved the submitted version.

*Competing interests.* The authors declare that they have no conflicts of interest.

*Acknowledgments*. We thank the Landesvermessungsamt Tirol for providing some of the aerial imagery. The comments of Etienne Berthier, Ben Pelto and one anonymous referee helped to substantially improve the paper.

*Financial Support*. This work was conducted as a part of the AlpSenseBench Project (2018-2019) funded by the Bavarian Ministry of Economic Affairs, Regional Development, and Energy and the AlpSenseRely Project (2020-2023) funded by the Bavarian State Ministry of the Environment and Consumer Protection.

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
