# Peer review of "Analyzing glacier retreat and mass balances using aerial and UAV photogrammetry in the Ötztal Alps, Austria"

_The Cryosphere, 2020_

## Referee Comment (RC1) · Ben Pelto (Referee) · 15 Dec 2020

**Review of Geissler et al. The potentials of high-resolution photogrammetry for analyzing glacier retreat in the Ötztal Alps, Austria**

Geissler et al. present a photogrammetric study using modern aerial photographs of the Vernagtferner and other glaciers in the Ötzal Alps. They calculate geodetic glacier mass balance for three periods between 2009–2018 from these photographs. They then compare these results with existing glaciological data for Vernagtferner and find a good overall comparison. The authors employ a UAV survey in 2018 to measure height change between the photographs and the glaciological observations, and produce a correction curve from these data. This robust method would be best served were it related to local meteorological data so that it could be employed more convincingly for the other two periods, and thus offer a pathway to other studies which commonly face the same temporal challenge of using photogrammetry to compare with glaciological data. The geodetic mass balance and associated error analysis is well done, but the study is lacking in depth in a few areas. The calculation of ice dynamics, here emergence and submergence, is insufficient and likely erroneous. A number of other points are introduced with insufficient detail, such as debris cover. A small extra effort on some of these points will greatly strengthen the paper. This manuscript will benefit from additional citations, and careful proofreading. Overall, I believe that this study is compelling and represents a valuable addition to The Cryosphere in both data and methodology, but requires further efforts to maximize the impact and relevance of the study. While I have numerous comments on the paper, I do not believe addressing these points will take a major investment of time.

**Major comments:**

Discussing emergence and submergence needs to be handled with caution absent stake observations of emergence and submergence, or model estimates of emergence and submergence. Given the magnitude of emergence and submergence, which is generally less than 0.5 meters, and the uncertainty in glaciological and geodetic mass balance, determining a change in, or even magnitude of vertical ice velocity is questionable. Further, as you indicate, the elevation of maximum volume loss, the rate of mass loss has increased and the ELA has increased. All three of these factors would trend towards a higher elevation where submergence occurs. Lacking more specific data or a more rigorous approach, I suggest caution in drawing conclusions from your estimates of emergence and submergence velocity.

Are GPS surveys conducted for the ablation stakes? If so there are a few methods from which you can estimate emergence and submergence velocities (Beedle Vincent 2020). If such data exist, then they must be incorporated here.

The coregistration procedure is not well described. In L159-160 you state that "The horizontal shift lies between 10 and 20 cm depending on the acquisition year and thus within the ground resolution of the images". Is this the pre-coregistration horizontal shift? This also sounds a bit small, if this is pre-coregistration, that's excellent. You also state that "Based on this mean vertical shift over stable ground, all DSMs except for the reference DSM were adjusted in height relative to the reference DSM of 2015." Does this mean that the coregistration was only vertical? Robust coregistration algorithms now exist to implement the method detailed in Nuth and Kääb

(2011). Should this be tested? This method removes not only vertical but also horizontal and rotational bias. Your Figure 6 and section 5.1.2 detail these errors well. Perhaps this is enough, I'm just curious why a full coregistration wasn't used, but having the error well described is sufficient.

For your altitude-related density function, additional explanation is required. This sounds like a good idea, but the particulars aren't clear enough. Over the ablation zone is the density held at 900 kg m3? Or does the density start to change prior to the equilibrium line? Klug et al. (2018) mapped snow/firn as one unit and ice as one unit and assigned a density to each. Pelto et al. (2019) mapped snow, firn and ice separately and assigned a density to each. If I'm reading this correctly, your function is only applied over the equilibrium line, i.e. holding density at 900 for the ice area, and 550 for snow, but using the linear function around the equilibrium line. This is unclear. If so, I think this an excellent approach. Also, does your method take into account the annual (or average) ELA position during each interval or a fixed ELA for the entire period?

Section 4.2. Your correction method is robust for 2018, nice Figure 3. I wonder whether a degree day function could be employed to reproduce the melt you observe in 2018, and then apply that function to the other two periods to adjust or produce a curve just like in 2018? Perhaps too much work for the small adjustment, but might be simple if there is some local temperature data. The correction method is one of the main selling points of your manuscript. I would suggest exploring a simple DDF or similar approach. If it proves reliable, this would greatly improve the applicability of your results. As you mention, using photogrammetric surveys to assess glaciological mass balance is challenging, because of time differences. By providing a simple framework to apply a present-day UAV survey to other time periods where none exist would be of great value and interest to the community (at least for relatively modern air photos).

The discussion is too wordy and redundant. Some sections could be combined and streamlined. Too often the discussion is restating the results section. The discussion should then better discuss questions raised by the reviewers and other under explained details.

Not enough references for previous photogrammetric work are given. Ensure that this is properly referenced and discussed in the intro and methods sections. Some suggested literature include (Baltsavias et al., 2001; Etzelmüller et al., 1993; Gudmundsson & Bauder, 1999; Magnússon et al., 2016; Nolan et al., 2015; Redpath et al., 2018)

**Specific comments:**

L11 Perhaps change to "are experiencing increasing mass loss...".

L14-15 Perhaps change "a significant glacier area" to something like "a heavily glacierized area".

L11-29 There is a mix of past and present tense in the abstract. Ensure you stick to one tense here.

L25-27 Awkwardly worded. Be clear that you find that the geodetic data can detect local details and deviations better.

42 Comma after geodetic is unnecessary.

45 Perhaps swap in "details" for demonstrations.

54 Start the sentence with "By combining" or move "This study presents" to the start of the sentence and end with "allowing the extraction..."

L69 Ensure consistency with numbers, here a dot is used, other times the number is presented without a dot (e.g. L77).

L105 Please explain what the two numbers in overlap mean in the caption.

L109 Change to "have been acquired...".

L112 The meters ice equivalent, or meters water equivalent? I'm not familiar with ice per water equivalent.

L128 Any citation for this at Vernagtferner? This has certainly been observed at many glaciers, but a couple citations here would be of value.

L131 Only up to 300-400 mm in the accumulation area? Can this be determined with only 4-5 measurements per year for what should be a 1-3 km2 accumulation area?

L132 Add parentheses around (2013). Ensure in-line references match TC style guide.

L176 Remove "additionally".

L177 Remove "as well".

L188 Define "SD".

L240 Nice figure. Add a scale bar?

L245 Nice figure. Add a scale bar to one of the panels?

L278-280 Why compare two different elevation bins?

L278-280 Should "Million" be lowercase? Change all instances if so.

L313-315 Suggest changing to "... , neglecting debris-covered areas within the glaciological interpolation led to an...".

L313-315 How was this debris cover value determined? (0.1m± 0.08w.e.a-1(0.8±0.6%). I suggest adding a few lines to the results and discussion on debris cover to better detail what you found. E.g. "...debris covered area experiences x.xx m of ablation on average versus x.xx m of proximal ice. This suggests that xxxx."

L325 Shorten the y-axis label and correct spelling and capitalization errors. Move x axis elevation labels to the top or bottom away from data points.

L352 Change to "quadrupled".

L349-L361 How do your results from these two periods compare with other regional estimates of mass balance?

L359 Exceed? It's hard to keep track of your comparisons of glaciological and geodetic mass balances. I recommend being explicit, rather than "greater than" "exceed", "lower", "higher" etc., use more positive or more negative, or express as more mass loss vs. less mass loss. Or if you stick with greater, lesser, etc, be sure to explain what each means here and use the same terms to discuss throughout. I found myself looking at figures and re-reading sections often to determine which method measured greater mass loss over given altitudes or time periods.

L376 Cut superfluous language. Here remove "As a result, for instance," and start the sentence with "We were able...".

L382 Why not calculate the height change for this dead ice body as an example if you're going to mention it here? Nice little advertisement of the detail and value of high resolution digital photogrammetry. You could even compare this to the mass loss on the toe of the glacier if desired.

L392 Insert "the" to make it "the SD".

L433-435 Useless? I think not, and you have just proved that they can be used, provided a correction. Instead end with something like "therefore require correction using geodetic survey data or other methods".

L469-470 I'm not sure of the value of this line, or why this is relevant here. Did the study at all address this topic?

**References**

Baltsavias, E. P., Favey, E., Bauder, A., Bosch, H., & Pateraki, M. (2001). Digital Surface

Modelling by Airborne Laser Scanning and Digital Photogrammetry for Glacier Monitoring.

*The Photogrammetric Record*, *17*(98), 243–273. https://doi.org/10.1111/0031-868X.00182

Etzelmüller, B., Vatne, G., Ødegârd, R. S., & Sollid, J. L. (1993). Mass balance and changes of surface slope, crevasse and flow pattern of Erikbreen, northern Spitsbergen: an application of a geographical information system (GIS). *Polar Research*, *12*(2), 131–146. https://doi.org/10.3402/polar.v12i2.6709

Gudmundsson, G. H., & Bauder, A. (1999). Towards an Indirect Determination of the Mass-balance Distribution of Glaciers using the Kinematic Boundary Condition. *Geografiska Annaler*, *81*(4), 575–583. https://doi.org/10.1111/j.0435-3676.1999.00085.x

Klug, C., Bollmann, E., Galos, S. P., Nicholson, L., Prinz, R., Rieg, L., Sailer, R., Stötter, J., & Kaser, G. (2018). Geodetic reanalysis of annual glaciological mass balances (2001–2011) of Hintereisferner, Austria. *The Cryosphere*, *12*(3), 833–849. https://doi.org/10.5194/tc-12-833-2018

Magnússon, E., Muñoz-Cobo Belart, J., Pálsson, F., Ágústsson, H., & Crochet, P. (2016). *Geodetic mass balance record with rigorous uncertainty estimates deduced from aerial photographs and lidar data–Case study from Drangajökull ice cap, NW Iceland*.

Nolan, M., Larsen, C., & Sturm, M. (2015). Mapping snow depth from manned aircraft on landscape scales at centimeter resolution using structure-from-motion photogrammetry. *The Cryosphere*, *9*(4), 1445–1463. https://doi.org/10.5194/tc-9-1445-2015

Pelto, B. M., Menounos, B., & Marshall, S. J. (2019). Multi-year evaluation of airborne geodetic surveys to estimate seasonal mass balance, Columbia and Rocky Mountains, Canada. *The Cryosphere*, *13*(6), 1709–1727. https://doi.org/10.5194/tc-13-1709-2019

Redpath, T. A. N., Sirguey, P., & Cullen, N. J. (2018). Mapping snow depth at very high spatial resolution with RPAS photogrammetry. *The Cryosphere*, *12*, 3477–3497. https://d-nb.info/1160680728/34

---

## Author Comment (AC1) · 6 Jan 2021

Dear referee, we highly appreciate your positive and constructive feedback. Indeed, most of your proposed changes of the manuscript are implementable and will not take too much time. We agree, that those small corrections will improve our manuscript and would therefore be happy to implement the following changes of the manuscript.

(A) We will relate the temporal correction function to local meteorological data. The respective data sets are already available to us. Following a DDF or similar approach, we will try to integrate this data to our correction function. This might slightly change our results but improve transferability.

[Figure]

(B) We will underline the advantages and successful implementation of our density function that includes a linear transition zone around the ELA. By doing so, we will eliminate ambiguities and clarify the methodology. For better understanding, you can find a plot of our density function below.

(C) We agree that the discussion of ice dynamics must be more careful since its values are similar to the magnitude of error. We will adjust the wording accordingly. However, we see great potential of applying our methodology to glaciers with higher ice dynamics and would like to present our quantitative results. GPS coordinates exist for the ablation stakes and we will check methods to estimate ice dynamics and compare them with our results.

(D) Concerning debris cover, we will add a few lines to the result and discussion as suggested. Since no ground truth data is available, we are not able to relate our findings to e.g. thickness or stone type.

(E) The DEM and orthophotos were coregistered horizontally by using ground control points. The resulting horizontal error is, as you mention, relatively small so that no further (rotational) coregistration was applied. We will clarify the method and will discuss, that a rotational coregistration might have improved observed errors (Fig. 6c).

(F) We will improve our introduction by incorporating previous photogrammetric work.

(G) We will improve our discussion by streamlining and incorporating questions that will come up during the discussion phase.

If you have any further questions or need a more detailed feedback, please let us know. Your specific comments will, of cause, be incorporated.

Sincerly, Joschka Geissler and Coauthors
* * *
[Figure]

**Fig. 1.** Density_function

---

## Referee Comment (RC2) · Anonymous Referee #2 · 12 Jan 2021

This paper uses results from photogrammetry for a small glacier region in the Ötztaler Alps. Using photogrammetry to study changes in glaciers is a popular method that can give good insights. The geodetic method has been conducted on many glaciers using various sources of data and often used for reanalysis of glaciological mass balance series.

The study would benefit from improving the explanations, figures and tables. More references could have been made to existing literature on similar studies (both on elevation changes and geodetic mass balance assessments). The manuscript would have benefitted from clearer writing, it is sometimes difficult to understand what the authors

mean.

Below I give some examples to exemplify what I mean as specific comments. Specific comments:

On the title: is 'the potentials of high-resolution photogrammetry' only relevant for the Otztal Alps? The title might imply so. I would change it: e.g. increased detail of glacier retreat and mass balance using high-resolution photogrammetry in the Ötztal Alps, Austria.

Abstract: focus on the findings. The abstract needs more polishing and be clearer. The first two sentences can be deleted. In the intro you also focus on the (European) Alps.

L14: take out significant, not needed (and what does it really mean, best to avoid).

L16, sentence starting glacier retreat. . .could be changes to: Glacier retreat, extent and surface elevation changes were analyzed for the 25 glaciers in the region, including Vernagtferner. Digital surface models (DSM) were generated from . . .. (Use direct language if possible. )

Change ' a correction was established' we used . . . to apply a correction Remove part 'which reveals the potential for a combination' & continue with the next sentence 'Results revealed . . .. ' L25 Could be -> were

L32. Add European before Alps. Could add 'hereafter referred to as the Alps' if this is the preference of the authors. Could also add Beniston et al. 2018.

Line 47. Here it would be nice to add reference to studies on the glacier, e.g. add 'e.g Escher-Vetter et al, 2009.'

Line 48. Here some refs could be added on various methods, the typical method used today by many mass balance investigators are laserscanning. See also later comment. e.g. Belart et al 2020, there are many studies on glacier changes using various methods and there are no references in this section.

L54. It is difficult to assess whether this method is unique when you provide so few references, what is unique about it? Repeated aerial photos and UAC are used in many studies.

L56. Why not just write: Suitable data for photogrammetric processing were available from 2009, 1015 and 2018, covering a period of 9 years.

The last part of chapter 1 from line 63 should be rewritten, see also comment to line 54.

L75. Could also here add reference to Esche-Vetter et al 2009 or other relevant literature on the glaciological mass balance work.

L76. can add map year in parentheses after Today (map year 20xx). Do you really measure the density of the firn, e.g. going past this year's summer surface? Usually, only snow pits are used for this year's mass balance. Measuring the winter snow in spring or remaining snow in autumn.

Figure 1. Instead of (Esri et al, 2020): Image source: ESRI (2020). The frame of lower left figure is partly visible.

Decimal separator: change ',' in figures and tables to dot '.' Yields throughout.

Chap 3.1. The first paragraph sounds like introduction and could be merged there.

L97. Use -> used In general in the paper: Use past tense where you describe work done by the authors for the paper, use present tense on published work.

Change title of 3.2 to Glaciological mass balance data??

L109. First sentence already mentioned before. Remove there or here.

L110-. At the same time ->which time. Be specific so we understand what you mean.

L112. Here you mention firn pits but on line 125 only snow pits. Is is not converting to water equivalent you mean? How is glaciological mass balance interpolated from

measurement date to fixed date? Did you use the outlines from 2015 also to reanalyze the mass balance?

Line 130/131. On mm what is the unit? Not water equivalents? What do you mean by error of the raster, you mean mass balance uncertainty in each grid cell? In this section references to work could also been added.

L136-140 Use past tense on work done for this study/where you describe the work you did. We use-> used (use past tense) is->was, are->were etc

L144. Fig.2. Full workflow (remove photogrammetric) because it involves more than that. In box instead of 'after huss et al 2013' you can rather write 'using fixed density factor

L149 (GNSS) -> using GNSS

L153 is agisoft not state of the art, maybe take-out state of the art in line 152

L148/L155: Is the ->was

LP6, last paragraph. Can mention the range of the adjustments.

L171/172. Is-> was

L173. More details of what? be specific. Of the xxx method or procedure or following. E.g. More details of the xx procedure/method can be found in xxx (year).

L176. Remove additionally. L 177. Remove as well.

L177. The details in this gradual change are unclear to me and firn density can be higher than 550. Suggest to rewrite it.

L186. Write out standard deviation SD first time mentioned.

L188.L192. Do you mean you use a fixed correlation based on finding in one year? But this will differ from year to year. Why not use meteorological data to estimate mass balance? This is a common approach. You could compare this to model estimates.

L240. Add in meter (m) after differences. I miss scale bare and glacier outlines. Add name Hintereisferner. There are several glaciers on the map.

L231. In general, glaciers have thinned and reduced in size.

Fig. 5. Outlines not velar/easy to read. Lacks scale bar and legend. Add name(s).

Fig 6. Readability of the figure could be improved to be clearer and sharper. Avoid grey background. Use black font. Replace ; with . in figure text.

Fig 7. Maybe it would be better to just show surface elevation changes or changes before and after correction.

L 288 add (geodetic) after photogrammetric.

Regarding this comparison, has a proper reanalysis f the data been carried out (ref. Zemp et al., 2013) for instructions. Have the outlines of 2015 been used in the mass balance calculations? This could be discussed.

L291: is-> was

L300. Is this tested against the procedure in Zemp et al (2013)? Should be referred to in this paragraph (the reference is in the paper but not referred to or used here).

317 do not only…., something is missing. Do you mean to connect the next sentence? Then rephrase.

318. is there a difference between photogrammetric and geodetic since, why not use geodetic?

Fig 9 it is not unproblematic to compare it that directly for shorter time periods due to submergence/emergence and density issues.

Figure 10. instead of a-e why not use a shortening of the glacier names?

L353. I suggest to also add the full period, could be good to have results for full period, better for density conversion as well.

Figure 11. here – could show the results spatially on a map, easier to see. I miss the names on location figure 1 or in the other figures.

L374/375. The potential for using photogrammetry such as this has been shown in multiple studies for a long time, and the authors should cite such studies. This is not novel. Would rewrite

L383. The dead ice body is not indicated in figure 4. Should be marked with a letter or number to help the reader.

L387-388. See comment to line 374/375- This is already stated. Again, it is not new to use photogrammetry to study glacier changes. I miss also a comparison with lidar studies, pros and cons using aerial photogrammetry versus lidar. Could be a good addition.

L403. Could use for the full period.

L409. Again, this applies only for that year and not the two other years. Here you should justify it by comparing met data and mass balance conditions for the three mapping years/dates. You mention it, but you are not using it. It must be meteorological data you could check and refer to.

L420. I am not sure it is representative for more than that year and that glacier. See comment above.

L433. What do you mean with great importance for future studies?

L452. Do you find that this is a proper reanalysis according to Zemp et al 2013 or should this be conducted, this is unclear to me. Here you can refer to 'reanalysis' and how others have used such data, there are several papers in the cryosphere (and other journals) on this topic. just search 'reanalysis'.

Data availability: The geodetic mass balance data is not available in a table in the paper. I suggest having a table in the paper that lists all the 25 glaciers with surface

elevation change, area of the mapping years and geodetic mass balance for the full period (9 yrs), and perhaps subperiods (e.g. in supplement). Adding a table in the paper gives users the chance to use the data further. Table could be accompanied with a figure in the paper showing the changes and glacier outlines. The tabular data could be submitted to WGMS, they store such data. In data availability thus add 'Geodetic mass balance data will be submitted to WGMS.'

References.

BELART, J., MAGNÚSSON, E., BERTHIER, E., PÁLSSON, F., AÃŘALGEIRSDÓTTIR, G., & JÓHANNESSON, T. (2019). The geodetic mass balance of Eyjafjallajökull ice cap for 1945–2014: Processing guidelines and relation to climate. Journal of Glaciology, 65(251), 395-409. doi:10.1017/jog.2019.16

Beniston, M and others (2018). The European mountain cryosphere: a review of its current state, trends, and future challenges, The Cryosphere, 12, 759–794, https://doi.org/10.5194/tc-12-759-2018, 2018.

Escher-Vetter, H., Kuhn, M., & Weber, M. (2009). Four decades of winter mass balance of Vernagtferner and Hintereisferner, Austria: Methodology and results. Annals of Glaciology, 50(50), 87-95. doi:10.3189/172756409787769672

---

## Author Comment (AC2) · 14 Jan 2021

Dear Referee,

thank you for your feedback, that is of great value for our manuscript and will further improve our publication.

In general, as already mentioned in our AC1,

- we will add more references to relevant publications in the introduction. More precisely, we will add references working on photogrammetry for glacier studies (RC1), elevation changes (RC2) and geodetic mass balance assessments (RC2).

[Figure]

- we will try to clarify explanations (e.g. gradual change of the density function) and the discussion.

- we will improve our results by incorporating meteorological data for the temporal correction, following the DDF or similar approach. This might slightly change our results but improve transferability.

We will take into account your specific comments while working on the revised manuscript. By doing so, we will

- improve some of our figures as suggested (e.g. thicken glacier outlines and add scalebar + legend (Fig. 5), mark dead ice body (Fig. 4), spatial visualisation of the geodetic mass balances (Fig. 11))

- add geodetic mass balances (9 yrs) of 25 glaciers to the supplement of this paper. This data will be submitted to WGMS.

We are convinced that those corrections and improvements will not take too long and will further improve the quality of our manuscript. Therefore, we would be happy to submit a revised manuscript after receiving the final response by the editor.

Joschka Geissler and Co-Authors

---

## Author Response (AR1)

Dear Ethienne, dear Referees,

thank you for your valuable feedback and specific comments. We revised our manuscript and were able to incorporate most of your comments. Our revised manuscript includes the following main improvements:

- Clarity and comprehensibility of figures, tables and explanations was improved. This was achieved by integrating the equations used, focussing on a consistent terminology (e.g. variable naming), and a more direct language.
- Our Introduction was restructured, is now more precise and more literature is cited.
- Our discussion was rewritten and restructured. The focus is set on the value of our findings for the community and on the potentials we see for future research.
- Meteorological data was integrated in our revision. This first of all improves our method since the transferability to other years where the correction was applied to is now much more robust. Moreover, our method can now be applied on other glaciers more accurately. In our opinion, the presented method of using a present-day UAV survey combined with meteorological data for the retrospective correction of geodetic mass balances is of great interest for the community.
- To compare glaciologic and geodetic mass balances a reanalysis following Zemp et al 2013 must be conducted, which we had included (reanalysis steps 1-4) in our previous manuscript versions. However, because our terminology partly differed from the original, the reader could not fully comprehend the reanalysis steps conducted. To address this issue, we adapted our terminology according to Zemp et al 2013 and directly refer to their reanalysis steps within our manuscript. This should support the interested reader to keep track of individual processing steps. It must be noted, as also stated in our manuscript and in the specific comments (Comment on Line 300), that reanalysis steps 5-6 were not performed, since systematic variations between both data sets were of main interest for the study.
- The results regarding "submergence and emergence" are based on strong assumptions and many potential error sources exist. However, we find that our results (Fig. 09 + 10) are interesting and of potential value for the community. We are now more careful with discussing those findings.

We again thank you for your feedback that strongly improved our results.

Sincerely,

Joschka Geissler + Coauthors.

| Line | Comment | Referee | Answer |
|---|---|---|---|
| 0 | Title: is 'the potentials of high-resolution photogrammetry' only relevant for the Otztal Alps? The title might imply so. I would change it: e.g. increased detail of glacier retreat and mass balance using high-resolution photogrammetry in the Ötztal Alps, Austria. | 2 | Thank you for your valuable suggestion, we discussed your feedback and adapted our title accordingly |
| 0 | Abstract: focus on the findings. The abstract needs more polishing and be clearer. The first two sentences can be deleted. In the intro you also focus on the (European) Alps. | 2 | We polised the abstract according to your feedback, the first sentences were deleted. |
| 11 | Perhaps change to "are experiencing i ncreasing mass l oss..." | 1 | First two sentences have been deleted |
| 11 | There is a mix of past and present tense i n the abstract. Ensure you stick to one tense here. | 1 | Applied |
| 14 | take out significant, not needed (and what does it really mean, best to avoid). | 2 | We now avoid the word "significant" since, as you mention, it was not used appropriately |
| 14 | Perhaps change "a significant glacier area" to something l ike "a heavily glacierized area" | 1 | We followed your suggestion and now call it a "heaviliy glacierized area" . See also comment above |
| 16 | sentence starting glacier retreat: : :could be changes to: Glacier retreat, extent and surface elevation changes were analyzed for the 25 glaciers in the region, including Vernagtferner. Digital surface models (DSM) were generated from : : :. (Use direct language if possible. ) Change ' a correction was established' we used : : : to apply a correction Remove part'which reveals the potential for a combination' & continue with the next sentence 'Results revealed : : :. ' | 2 | We improved our manuscript similar to your suggestions. We now use direct lanuage where possible. |
| 25 | Could be -> were | 2 | Applied |
| 25 | Awkwardly worded. Be clear that you find that the geodetic data can detect l ocal details and deviations better | 1 | We hope our new version is clearer and more targeted. |
| 32 | Add European before Alps. Could add 'hereafter referred to as the Alps' if this is the preference of the authors. Could also add Beniston et al. 2018. | 2 | We followed your suggestion and added "European Alps, hereafter referred to as the Alps". We added your suggested reference. |
| 42 | Comma after geodetic i s unnecessary. | 1 | Applied |
| 47 | . Here it would be nice to add reference to studies on the glacier, e.g. add 'e.g Escher-Vetter et al, 2009.' | 2 | Applied |
| 48 | Here some refs could be added on various methods, the typical method used today by many mass balance investigators are laserscanning. See also later comment. e.g. Belart et al 2020, there are many studies on glacier changes using various methods and there are no references in this section. | 2 | On your advice, we restructured our introduction. We now mention ALS and refer to Baltsavias 2001 for a comparison of both geodetic methods. We give many other (photogrammetric) references (L49ff) |
| 54 | Start the sentence with "By combining" or move "This study presents" to the start of the sentence and end with "allowing the extraction..." | 1 | Applied |
| 54 | It is difficult to assess whether this method is unique when you provide so few references, what is unique about it? Repeated aerial photos and UAC are used in many studies. | 2 | We are now more specific what is new about our method (L59): Using an UAV survey and meteoroloical data for retrospective correction of geodetic acquisition dates. |
| 56 | Why not just write: Suitable data for photogrammetric processing were available from 2009, 1015 and 2018, covering a period of 9 years. The last part of chapter 1 from line 63 should be rewritten, see also comment to line 54. | 2 | We restructured to Introduction. The last part was rewritten according to your advice. |
| 69 | Ensure consistency with numbers, here a dot is used, other times the number i s presented without a dot (e.g. L77). | 1 | We checked our manuscript accordingly. |
| 75 | Could also here add reference to Esche-Vetter et al 2009 or other relevant literature on the glaciological mass balance work. | 2 | L73; We added the proposed reference. |
| 76 | can add map year in parentheses after Today (map year 20xx). | 2 | Thank you for pointing out this bad wording, corrected! (L74) |
| 97 | Use -> used In general in the paper: Use past tense where you describe work done by the authors for the paper, use present tense on published work. Change title of 3.2 to Glaciological mass balance data?? | 2 | We are now consistend with the use of past/present tense. We changed the title if 3.2. according to your suggestion. |
| 105 | Please explain what the two numbers i n overlap mean i n the caption. | 1 | Applied; See Table 1 header (L95) |
| 109 | Change to "have been acquired...". | 1 | Sentence one and two have been merged |
| 109 | First sentence already mentioned before. Remove there or here. | 2 | Sentence one and two have been merged |
| 110 | At the same time ->which time. Be specific so we understand what you mean. | 2 | Clearified. |
| 112 | The meters i ce equivalent, or meters water equivalent? I'm not familiar with i ce per water equivalent. | 1 | This was a typo, corrected |

| | | | |
|---|---|---|---|
| 112 | i) Here you mention firn pits but on line 125 only snow pits; Do you really measure the density of the firn, e.g. going past this year's summer surface? Usually, only snow pits are used for this year's mass balance. Measuring the winter snow in spring or remaining snow in autumn.
ii) How is glaciological mass balance interpolated from measurement date to fixed date?
iii) Did you use the outlines from 2015 also to reanalyze the mass balance? | 2 | i) Since density is measured in snow and firn pits, we added "firn" to L125, even though firn is only measured if necessary (e.g. in the case of determining the last accumulation layer)
ii) The time difference is usually only a few days between the measurements and the fixed date. For these days we use information from our automatic stations on the glacier to correct for additional melt or accumulation, by scaling the mass balance field according to the data from the automatic stations.
iii) We use annual glacier outlines at the beginning of the mass balance period for computing glaciological mass balance. |
| 128 | Any citation for this at Vernagtferner? This has certainly been observed at many glaciers, but a couple citations here would be of value. | 1 | Glaciologic data of the Vernagtferner was used within Zemp et al. (2013). This reference is given in the manuscript. |
| 130 | On mm what is the unit? Not water equivalents? What do you mean by error of the raster, you mean mass balance uncertainty in each grid cell? In this section references to work could also been added. | 2 | i) changed unit to m w.e.
ii) clearified. This is the mean error of the interpolated glaciologic raster.
iii) The data and error used within this study is part of the study Zemp et al. (2013). This reference is given in the manuscript. |
| 131 | Only up to 300-400 mm i n the accumulation area? Can this be determined with only 4-5 measurements per year for what should be a 1-3 km2 accumulation area? | 1 | Yes, from manual measurements in spring we have information on the snow-& firn distribution within the accumulation area. By integrating this information within the glaciologic interpolation, the number of stakereadings is sufficient. Our results (Fig. 10), thus the comparison with geodetic data, underline the quality of our interpolation. |
| 132 | Add parentheses around (2013). Ensure i n-line references match TC style guide. | 1 | Checked |
| 136 | Use past tense on work done for this study/where you describe the work you did. We use-> used (use past tense) is->was, are->were etc | 2 | See comment on Line 97 |
| 144 | Fig.2. Full workflow (remove photogrammetric) because it involves more than that. In box instead of 'after huss et al 2013' you can rather write 'using fixed density factor | 2 | Thank you for your feedback. We corrected the figure accordingly. This figure was further improved, going beyond your comment. |
| 148 | Is the ->was | 2 | Changed |
| 149 | (GNSS) -> using GNSS | 2 | L138, applied |
| 153 | is agisoft not state of the art, maybe take-out state of the art in line 152 | 2 | We took out state of the art. |
| 171 | Is-> was | 2 | Applied |
| 173 | More details of what? be specific. Of the xxx method or procedure or following.E.g. More details of the xx procedure/method can be found in xxx (year). | 2 | Applied |
| 176 | Remove "additionally". | 1 | Applied |
| 177 | Remove "as well". | 1 | Applied |
| 177 | The details in this gradual change are unclear to me and firn density can be higher than 550. Suggest to rewrite it. | 2 | We now provide the equation used. This part of the manuscript was clearified and rewritten. L170ff |
| 186 | Write out standard deviation SD first time mentioned. | 2 | Applied |
| 188 | Define "SD". | 1 | Applied, L187 |
| 192 | Do you mean you use a fixed correlation based on finding in one year? But this will differ from year to year. Why not use meteorological data to estimate mass balance? This is a common approach. You could compare this to model estimates. | 2 | Thank you for this feedback. The former version of our correction was improved by incorporating meteorological data. Sect. 4.2. discribes the new method.
We thus use the UAV survey to derive the spatial pattern of our correction and meteorological data for taking into account positive degree day sums of the different periods to be corrected. |
| 231 | In general, glaciers have thinned and reduced in size. | 2 | Applied |
| 240 | Nice figure. Add a scale bar? | 1 | In 3D View, no scale bar can be added. |
| 240 | Add in meter (m) after differences. I miss scale bare and glacier outlines. Add name Hintereisferner. There are several glaciers on the map. | 2 | In 3D View, no scale bar can be added; We added the name and outlie of the Hintereisferner |
| 245 | Nice figure. Add a scale bar to one of the panels? | 1 | In 3D View, no scale bar can be added. |

| | | | |
|---|---|---|---|
| 278 | Why compare two different elevation bins? | 1 | We compare the elevation bins of the maximum volume loss. Since the elevation of the maximum volume loss increased, we compare two different elevation bins. We clearified the phrasing. |
| 278 | Should "Million" be l owercase? Change all i nstances i f so. | 1 | Applied |
| 288 | add (geodetic) after photogrammetric.Regarding this comparison, has a proper reanalysis f the data been carried out (ref. Zemp et al., 2013) for instructions. Have the outlines of 2015 been used in the mass balance calculations? This could be discussed. | 2 | i) For Reanalysis Zemp see Comment to L300
ii) The outlines of 2015 have been used for mass balance calculations, following Fischer et al. (2015). We now provide formulas used: See Eq. 2; |
| 291 | is-> was | 2 | Applied |
| 300 | Is this tested against the procedure in Zemp et al (2013)? Should be referred to in this paragraph (the reference is in the paper but not referred to or used here). | 2 | We applied Reanalysing Steps 1-4 according to Zemp et. al (2013). Systematic and random error of the geodetic error was derived following Nuth and Kääb (2011). Systematic and random error of glaciologic data was assumed to be 0.1 m w.e. a^-1. This is according to Zemo et. al 2013 since the glaciologic data was used within Zemp et. al (2013). Reanalysing steps 5+6 (Zemp et. al 2013) were not applied, since the systematic error revealed by comparing geodetic and glaciologic data was of main interest for this study. This is now clearified within the manuscript. |
| 313 | Suggest changing to "... , neglecting debris-covered areas within the glaciological interpolation led to an..." | 1 | we added a few lines, going more into depth regarding debris cover - both in the results and in the discussion |
| 313 | How was this debris cover value determined? (0.1m± 0.08w.e.a-1(0.8±0.6%). I suggest adding a few l ines to the results and discussion on debris cover to better detail what you found. E.g. "...debris covered area experiences x.xx m of ablation on average versus x.xx m of proximal i ce. This suggests that xxxx." | 1 | Method is better described now in L213; FYI: We determine debris cover using the variation raster.; See also comment above |
| 317 | do not only: : :., something is missing. Do you mean to connect the next sentence? Then rephrase. | 2 | Rephrased |
| 318 | is there a difference between photogrammetric and geodetic since, why not use geodetic? | 2 | We understand photogrammetric as a subordinate term of geodetic, because it is possible to determine geodetic mass balances also with non-photogrammetric sensors (eg ALS). We improved this within our manuscript by using the word photogrammetric where we directly refer to our dataset (eg Sect. Data Acquisition). Whenever we write about mass balances, we use the word geodetic. This should improve the comprehensibility of our paper, since geodetic is the standard term used in the context of glacier mass balances. |
| 325 | Shorten the y-axis l abel and correct spelling and capitalization errors. Move x axis elevation l abels to the top or bottom away from data points. | 1 | Applied |
| 349 | How do your results from these two periods compare with other regional estimates of mass balance? | 1 | We now compare our result of the Hintereisferner with WGMS-data. The comparison was added to the discussion. L370ff |
| 352 | Change to "quadrupled". | 1 | Applied |
| 353 | I suggest to also add the full period, could be good to have results for full period,better for density conversion as well | 2 | We added the full period. The geodetic mass balance data of all glaciers and all periods is now provided in the Appendix A and will also be submitted to WGMS |
| 359 | Exceed? It's hard to keep track of your comparisons of glaciological and geodetic mass balances. I recommend being explicit, rather than "greater than" "exceed", "lower", "higher" etc., use more positive or more negative, or express as more mass l oss vs. l ess mass l oss. Or i f you stick with greater, l esser, etc, be sure to explain what each means here and use the same terms to discuss throughout. I found myself l ooking at figures and re-reading sections often to determine which method measured greater mass l oss over given altitudes or time periods. | 1 | We must admit that we were not consistent with the diction of our compariosons in our submitted manuscript. In our revised manuscript we are now more explicit by only using "more negative" or "less positive" etc... In our opinion, this improved the readability of our comparisons. |
| 374 | The potential for using photogrammetry such as this has been shown in multiple studies for a long time, and the authors should cite such studies. This is not novel. Would rewrite | 2 | This section was rewritten. L370ff |

| | | | |
|---|---|---|---|
| 376 | Cut superfluous language. Here remove "As a result, for instance," and start the sentence with "We were able…". | 1 | Applied |
| 382 | Why not calculate the height change for this dead ice body as an example if you're going to mention it here? Nice little advertisement of the detail and value of high resolution digital photogrammetry. You could even compare this to the mass loss on the toe of the glacier if desired. | 1 | Thank you for your nice idea. We derived the volume change for the Dead-Ice body and mention it within our discussion.  See L381 |
| 383 | The dead ice body is not indicated in figure 4. Should be marked with a letter or number to help the reader. | 2 | We added a letter to fig 4. See also comment above. |
| 387 | See comment to line 374/375- This is already stated. Again, it is not new to use photogrammetry to study glacier changes. I miss also a comparison with lidar studies, pros and cons using aerial photogrammetry versus lidar. Could be a good addition. | 2 | Thank you for your input. However, we decided that a comparison between the geodetic measurement methods (photogrammetric vs. LiDAR) should not be part of our paper. This would go beyond the scope of this paper and since no lidar data was available for this publication this comparison would only be therotical. Additionally, there are already publications that make such comparisons. (check Baltsavias et al., 2001) |
| 392 | Insert "the" to make it "the SD". | 1 | Applied |
| 403 | Could use for the full period. | 2 | Thank you for your comment that showed us, that this sentence had to be clearified. L393 |
| 409 | Again, this applies only for that year and not the two other years. Here you should justify it by comparing met data and mass balance conditions for the three mapping years/dates. You mention it, but you are not using it. It must be meteorological data you could check and refer to. | 2 | We now integrate meteorological data within our temporal correction. Thank you for pointing this out. See comment on Line 192 |
| 420 | I am not sure it is representative for more than that year and that glacier. See comment above. | 2 | Thank you for your feedback. We agree, that our correction function is only valid for the Vernagtferner.  This is why we suggest to conduct an UAV survey in order to get correction functions that are valid for other glaciers.
In our revised manuscript we are now including meteorological data. This improved method is now also valid for different years and periods. See comment on line 192 |
| 433 | What do you mean with great importance for future studies? | 2 | We are now more specific where we see the potential for future studies L446ff |
| 433 | Useless? I think not, and you have just proved that they can be used, provided a correction. Instead end with something like "therefore require correction using geodetic survey data or other methods". | 1 | This was a misunderstanding. See comment above. |
| 452 | Do you find that this is a proper reanalysis according to Zemp et al 2013 or should this be conducted, this is unclear to me. Here you can refer to 'reanalysis' and how others have used such data, there are several papers in the cryosphere (and other journals) on this topic. just search 'reanalysis'. | 2 | See comment on Line 300 |
| 469 | I'm not sure of the value of this line, or why this is relevant here. Did the study at all address this topic? | 1 | This line addresses the fact, that the UAV survey presented in this paper did not cover the entire glacier area. For future studies, we recommend covering the entire glacier. This line was rewritten. |
| Chap 3.1 | The first paragraph sounds like introduction and could be merged there. | 2 | Thank you for your feedback. We agree that this paragraph should not have been part of the Data Acquisition section. Merged to Introduction and Discussion |
| a availat | The geodetic mass balance data is not available in a table in the paper. I suggest having a table in the paper that lists all the 25 glaciers with surface elevation change, area of the mapping years and geodetic mass balance for the full period (9 yrs), and perhaps subperiods (e.g. in supplement). Adding a table in thepaper gives users the chance to use the data further. Table could be accompanied with a figure in the paper showing the changes and glacier outlines. The tabular data could be submitted to WGMS, they store such data. In data availability thus add 'Geodetic mass balance data will be submitted to WGMS.' | 2 | We now provide the areas, surface changes, and geodetic mass balances of all glaciers for all periods in the appendices . They will also be submitted to the WGMS. We changed the Data Availability section accordingly. |
| Fig 1 | Instead of (Esri et al, 2020): Image source: ESRI (2020).
The frame of lower left figure is partly visible. Decimal separator: change ',' in figures and tables to dot '.' Yields throughout | 2 | We changed the decimal seperator and the citation of the image source. In the new version, the frame of the lower right figure is now visible. |
| Fig 10 | instead of a-e why not use a shortening of the glacier names? | 2 | Applied, thank you for your advice that makes our figure more intuitive. |

| | | | |
|---|---|---|---|
| Fig 11 | here – could show the results spatially on a map, easier to see. I miss the names on location figure 1 or in the other figures. | 2 | Thank you for your feedback. We followed your suggestions by integrating a map of all glacier outlines in figure 11.
We also plotted the bars spatially on a map. However, this reduces the ability of the reader to compare the magnitudes of the geodetic mass balances since they are i) not aligned and ii) smaller. Thus, we decided to remain with our bar diagram. |
| Fig 5 | Outlines not velar/easy to read. Lacks scale bar and legend. Add name(s). | 2 | Thank you for your comment and your feedback. We considered your suggestions. Please note, that in 3D View, no scale bar can be added. We added a legend and thickened the outlines. |
| Fig 6 | Readability of the figure could be improved to be clearer and sharper. Avoid grey background. Use black font. Replace | 2 | We improved readability following your advices. |
| Fig 7 | Maybe it would be better to just show surface elevation changes or changes before and after correction. | 2 | Thank you for your feedback on this figure. We modified the plot on the top left, so that the reader can now see the surface changes of the Vernagtferner before and after the correction. For the volume change, we decided to remove data and only show the regression. This simplifies the figure and increases understandability. |
| Fig 9 | it is not unproblematic to compare it that directly for shorter time periods due to submergence/emergence and density issues. | 2 | We discussed your feedback on this figure. This comparison of glaciologic and geodetic mass balance data is of great importance for our paper, since
i) it shows that the glaciologic mass balance data and our correction has weaknesses within the accumulation area
ii) allows an estimate of dynamic processes (see fig. 10)
However, we are aware that other sources of error exist (e.g. density conversion). We discuss this within our discussion |
| | Discussing emergence and submergence needs to be handled with caution absent stake observations of emergence and submergence, or model estimates of emergence and submergence. Given the magnitude of emergence and submergence, which is generally less than 0.5 meters, and the uncertainty in glaciological and geodetic mass balance, determining a change in, or even magnitude of vertical ice velocity is questionable. Further, as you indicate, the elevation of maximum volume loss, the rate of mass loss has increased and the ELA has increased. All three of these factors would trend towards a higher elevation where submergence occurs. Lacking more specific data or a more rigorous approach, I suggest caution in drawing conclusions from your estimates of emergence and submergence velocity. Are GPS surveys conducted for the ablation stakes? If so there are a few methods from which you can estimate emergence and submergence velocities (Beedle Vincent 2020). If such data exist, then they must be incorporated here. | | Thank you for your feedback. We now discuss emergence and submergence with more caution. GPS-Data is currently not available for the Vernagtferner. |
| | The coregistration procedure is not well described. In L159-160 you state that "The horizontal shift lies between 10 and 20 cm depending on the acquisition year and thus within the ground resolution of the images". Is this the pre-coregistration horizontal shift? This also sounds a bit small, if this is pre-coregistration, that's excellent. You also state that "Based on this mean vertical shift over stable ground, all DSMs except for the reference DSM were adjusted in height relative to the reference DSM of 2015." Does this mean that the coregistration was only vertical? Robust coregistration algorithms now exist to implement the method detailed in Nuth and Kääb (2011). Should this be tested? This method removes not only vertical but also horizontal and rotational bias. Your Figure 6 and section 5.1.2 detail these errors well. Perhaps this is enough, I'm just curious why a full coregistration wasn't used, but having the error well described is sufficient. | | Thank you for your comment on the coregistration. Yes, this horizontal shift given is the "pre-coregistration horizontal shift". As you say, this value is excellent and we thus decided to only perform a vertical co-registration.
As you mention, we describe the remaining errors within our accuracy assessment. We also added a paragraph in discussion where we mention that the (admittedly small) rotational errors may be adressed if a horizontal and rotational coregistration would have been performed. L385ff |

| | |
|---|---|
| For your altitude-related density function, additional explanation is required. This sounds like a good idea, but the particulars aren't clear enough. Over the ablation zone is the density held at 900 kg m3? Or does the density start to change prior to the equilibrium line? Klug et al. (2018) mapped snow/firn as one unit and ice as one unit and assigned a density to each. Pelto et al. (2019) mapped snow, firn and ice separately and assigned a density to each. If I'm reading this correctly, your function is only applied over the equilibrium line, i.e. holding density at 900 for the ice area, and 550 for snow, but using the linear function around the equilibrium line. This is unclear. If so, I think this an excellent approach. Also, does your method take into account the annual (or average) ELA position during each interval or a fixed ELA for the entire period? | We clarified our altitude-relateted density function and provide the equation used. This should answer your questions. But for your information: You have correctly understood our density-function. The ELA used for the density function is the mean ELA altitude of the respective period. L168ff |
| Section 4.2. Your correction method is robust for 2018, nice Figure 3. I wonder whether a degree day function could be employed to reproduce the melt you observe in 2018, and then apply that function to the other two periods to adjust or produce a curve just like in 2018? Perhaps too much work for the small adjustment, but might be simple if there is some local temperature data. The correction method is one of the main selling points of your manuscript. I would suggest exploring a simple DDF or similar approach. If it proves reliable, this would greatly improve the applicability of your results. As you mention, using photogrammetric surveys to assess glaciological mass balance is challenging, because of time differences. By providing a simple framework to apply a present-day UAV survey to other time periods where none exist would be of great value and interest to the community (at least for relatively modern air photos). | Thank you for this helpful comment. We now apply a simple DDF approach in the context of our correction. L194 - L205 By doing so we improved not only the transferabilty of our method but also the results itself. As you say, the relatively simple method presented in this paper for using present-day UAV surveys for retrospective correction of geodetic mass balance data is of great value to the community. We underline this also within our discussion. L399-L419 |
| The discussion is too wordy and redundant. Some sections could be combined and streamlined.Too often the discussion is restating the results section. The discussion should then better discuss questions raised by the reviewers and other under explained details | The discussion was completely rewritten and structured. We use more direct language and thus reduce superfluous language. |

| | | | |
|---|---|---|---|
| The study would benefit from improving the explanations… | 2 | Thank you for your feedback. Following the specific comments, we improved readability, understandability of our explanations, figures and tables. In general, for an easier understanding, we are consistent with our choice of colours: Blue Period 15-18, Red Period 09-15; Orange Period 09-18. We checked all our figures so that they follow the terminology of the manuscript. |
| …figures | 2 | Fig 1: We changed the decimal seperator and the citation of the image source. In the new version, the frame of the lower right figure is now visible. Fig 2: This figure was modified, since the integration of meteorological data changed our methodogy Fig 3: Axis and equation were renamed so that they match with the equations in the manuscript. A black outline improves the quality of this figure. Fig 4: We added the names of the highest summit and the name and outline of the Hintereisferner. Fig 5: We thickend the outlines, changed colours, added a legend and added the names of the glacier as well as a summit. This clearly improves visibility and understandability of this figure. Fig 6: The revised figure is now clearer and sharper. We avoid grey background and use black font. Fig 7: We changed the axis names following the terminology of the manuscript. Added corrected geodetic data for the Vernagtferner. Line style and colour are match throughout all four plots: Dotted: Corrected geodetic data, blue: period 15-18, red: period 09-15 Fig 8: We changed the axis names following the terminology of the manuscript. Fig 9: We changed the axis names following the terminology of the manuscript. Fig 10: Instead of using a-e we now use abbriviations of the accumulation area names following the specific comments.. This increases understandability. Fig 11: Here we added a map of all glaciers within the study area. This allows the reader to visually link geodetic mass balances and glacier extend, location etc and increases the understanding of our study area. |
| …and tables | 2 | Table 1: We added the units in the header. Explanation is given on the overlap. Table 2: This table was completely revised including the new terminology of the manuscript. This improves understandability. |
| More references could have been made to existing literature on similar studies (both on elevation changes and geodetic mass balance assessments). | | We added the following literature on elevation changes and geodetic mass balance assessments. Belart et al., 2019; Jaenicke et al., 2006; Magnússon et al., 2016; Mayer et al., 2017, Gudmundsson and Bauder, 1999 |
| The manuscript would have benefited from clearer writing, it is sometimes difficult to understand what the authors mean | | We have completely revised our manuscript and tried to improve its comprehensibility. For instance, see comments on Line 105, 177, 313, 318, 359 |

---

## Referee Report (RR1)

The corrections and changes made by the authors have greatly strengthened this manuscript. The manuscript reads well now and key gaps have been satisfactorily addressed. In particular, the uncertainty analysis has been addressed and discussed and the transferability of the method improved. The approach of using a melt model to inform the correction of Ba from photogrammetric DSMs for years without UAV surveys should prove a valuable contribution to the glaciological community. I believe that the manuscript can be published as is and leave it to the discretion of the editor and other referee whether any minor changes should be made.

Minor points

L215 Change "intense glaciological surveys" to "detailed", "rigorous", or "thorough".

L543 Perhaps only recommend adjusting the density of firn, the density of ice is generally assumed to be unchanged between regions.

L648 I would avoid stating that this is the main disadvantage of photogrammetry since the associated biases and errors were not investigated here, get right to the point.

---

## Author Response (AR2)

Dear Etienne Berthier,

thank you very much for your feedback.

We are very sorry that referee #2 did not have the clean version of our manuscript and we understand very well that only with the tracked-changes document, editing must have been time-consuming and annoying. Unfortunately, we do not have a reason for this technical error and hope that this time everyone will have all the revised versions of the manuscript.

During this revision period, we have formulated our point-by-point responses in more detail and separately for each referee.

We intensively discussed all the feedback we received from the two referees and you. Besides minor changes, we revised the introduction and discussion in order to contextualize our methods and results and hope that those changes meet your expectations.

All co-authors and two independent readers proof-read the manuscript.

If any further questions arise, feel free to contact us. Looking very much forward to your final decision.

Kind regards,

Joschka Geissler and Co-Authors

*Dear Ben Pelto,*

*your positive feedback on our revised manuscript is highly appreciated. The improvements of the presented method (e.g. transferability) and also textual quality were only possible thanks to your valuable feedback during the interactive discussion. We agree that those changes greatly improved our manuscript itself and increased its benefit for the scientific community. In our now published version of the manuscript, we have further revised and improved the contextualization in the scientific context and our conclusion to emphasize this benefit even more.*

*Apparently, there was a technical problem with the revision, so that referee #2 did not receive the clean version of the revised manuscript. However, we hope you received all uploaded versions, even though your line-numbers also correspond to the authors-tracked-changes-document.*

*Best regards,*

*Joschka Geissler and Co-Authors*

**Point-by-Point answer to your feedback**

**L215 Change "intense glaciological surveys" to "detailed", "rigorous", or "thorough".**

*Sentence has been rewritten. Now L197*

**L543 Perhaps only recommend adjusting the density of firn, the density of ice is generally assumed to be unchanged between regions.**

*Thank you for the good advice. This simplifies this recommendation. New (L450):*

*"We recommend, however, adjusting the firn density values to the corresponding region."*

**L648 I would avoid stating that this is the main disadvantage of photogrammetry since the associated biases and errors were not investigated here, get right to the point.**

*We now avoid this more general conclusion and are now more specific on where we see the potential of our correction. New (L436):*

*"This correction as well as our density function allow a spatially explicit, retrospective determination and correction of geodetic mass balances and is transferable to other glaciers. The results show that the glaciological method can be greatly complemented with photogrammetric analyses, increasing the accuracy of the glaciological mass balance series, revealing regions of anomalous mass balance conditions, and allowing estimates of the imbalance between mass balance and ice dynamics."*

Dear Referee,

thank you for your detailed review of our manuscript. If we understand you correctly, the clean version of our manuscript was not provided to you and we totally understand that doing a review only with the "authors tracked changes" was challenging. Considering the high number of changes made, this must have been annoying. We have no explanation why the clean version of the manuscript was not provided to you and are sorry for this inconvenience. We hope that you will receive all documents after this review. If not, we kindly ask you to refer to TC and ask for all uploaded documents.

Please note that our authors-tracked-changes document was created using the MS Word "Compare documents" tool. This is also suggested to use by TC: https://www.the-cryosphere.net/submission.html#manuscriptcomposition. Changing the resulting layout etc is – at least to my knowledge – not possible. For this review, we improved our authors-tracked-changes document considering your advice: References-changes are now highlighted and old versions of changed figures are now shown in black and white.

We also provide, as requested, a point-by-point answer to both referees separately.

In addition to your comments,

- the manuscript was proofread by all co-authors.
- all confusing sentences that we found were improved.
- we revised our introduction and parts discussion for a better contextualization of our work.

Concerning your comment on our Author Comments (AC1 and AC2), we just want to clarify that the authors are aware that a major revision takes its time. We incorporated most of the referee comments during the first revision of the manuscript and thus took the time needed. The authors again thank the referees and editor for their valuable feedback during the review process that greatly improved the quality of our publication.

Sincerely,

Joschka Geissler and Co-authors.

**Point-by-point-answer:**

**Title: The authors write: 'Thank you for your valuable suggestion, we discussed your feedback and adapted our title accordingly'. They did not adapt the title accordingly. The title is changed, but not as suggested, it is strange that they then write accordingly. Such an example of adaptation is confusing and not very convincing.**
*First of all, to explain this confusing comment: During the last revision, we have discussed your comment on our title intensively. We first decided to change our title according to your suggestions but ended up not to do so. Unfortunately, we forgot to change our comment accordingly. We are very sorry for this confusion.*

*We discussed this comment again and would like to adapt our title following your suggestions. We agree, that the new version of the title tends less for misinterpreting the title that our methods are only be applicable to the Ötztal. Additionally, this new title adds the word 'mass balances', that represent a major part of our results.*

*New title: Analyzing glacier retreat and mass balances using aerial and UAV photogrammetry in the Ötztal Alps, Austria.*

**Line 35: 'The availability of high -resolution multi-temporal digital aerial imagery for most of the glaciers in the Alps will provide a more comprehensive and detailed analysis of climate change- induced glacier retreat.' Is it really the availability of images that provide .. I would rather say 'the availability of …. images provide opportunities for a more comprehensive and detailed analysis of glacier retreat.'**
*L23: We agree with your suggestion and added 'opportunities'. New: The availability of (...) imagery (...) provides opportunities for a more comprehensive and detailed analysis of climate change induced glacier retreat.*

**Line 67: the authors here write on airborne laser scanning and digital photogrammetry as standard methods, but here one could also differ between data sources (aerial and satellite imagery and lidar) and methods. One can also combine various data sources to obtain geodetic mass balance so this section could been improved in my opinion**

*This section was improved and rewritten. New version:*

L 38: *"For retrieving geodetic mass balance data, different remote sensing methods exist, varying in the platform (e.g. satellite, airplane, UAV) and sensor (e.g. Laser Scanner, Optic Camera, Radar) used. Their specific benefits and limitations have been analyzed and discussed in different studies (Baltsavias et al., 2001; Bamber and Rivera, 2007; Kääb, 2005; Pellikka and Rees, 2010)."*

**Line 68. Present tense on published data, e.g. are, but here you could also add newer literature, this is 20 years old.**
*This sentence was completely revised. We proofread our manuscript and checked for tenses.*

**Line 70. But the methods can be combined so don't understand this reasoning, Belart et al (2019) combines several DEMs.**
*We completely revised our introduction and hope that we now clarified the contextualization of our manuscript. See also comment on line 82.*

**Line 77. It is common to apply a correction to compare geodetic and glaciological, this is explained in Zemp et al. (2013) and many other papers, e.g. Andreassen et al (2016) (https://tc.copernicus.org/articles/10/535/2016/)**
*L73: We completely revised our introduction and hope that we now clarified the contextualization of our manuscript. See also comment below.*

**Line 82-83. I still don't understand what is unique by this, using meteorological data for correction and using drone data is not new. See point above. Maybe it is just clumsy writing, but I expect more of the revised version.**
*L73: We completely revised our introduction and hope that we now clarified the contextualization of our manuscript. Geodetic mass balance extrapolation is commonly performed by e.g. using field data or meteorological data (e.g. using degree-day models). We use, as you know, an additional UAV-survey that provides us an altitude-dependent correction-function. By then combining our simple degree-day approach to this correction*

*function we receive an altitude-dependent degree-day function. This then allows an altitude-specific correction of our geodetic mass balances that is also applicable to 2015 and 2009, by incorporating the positive degree day sums of the respective correction periods. Such a combination of methods has, to our knowledge, not been used in earlier studies.*

**Line 84. Here you write this calibration but the line before does not talk about calibration.**
*We completely revised our introduction. This sentence does not exist anymore.*

**Line 91. Here you refer to glaciated, in the abstract you refer to glacierized. This is not the same. Glaciated is often referred to as covered by glaciers in the past.**
*L94: We changed this according to your suggestion to 'glacierized'.*

**Line 137. I would say the uncertainty can be higher than 1 cm for stake readings when surface is uneven (ice).**

*L 131: For the Vernagtferner, stake reading conditions are relatively good. (Ice)surface is relatively flat and thus the mean error is typically about 1 cm. However, this is the mean error and not the maximum, that can of cause be higher.*

**Line 153. Why separate ablation area and accumulation are, is it not ablation and accumulation on the entire glacier?**
*L144: We discussed this comment. Unfortunately, we do not completely understand what you mean. We think, your comment is on this sentence:*

*While ablation varies between 0 and up to 4.5 m w.e. a-1 in the ablation area, accumulation only varies between 0 and about 0.3-0.4 m w.e. a-1 in the accumulation area.*

*With this sentence, we want to show that the variability of the specific mass balance is higher within the ablation area then in the accumulation area.*

**Line 204. It is common to use average glacier area, e.g. Zemp et al (2013) that you refer to. This will impact the results and you need to recalculate it to compare with glaciological balance.**
*L190: Please note that our geodetic mass balances were derived following the exact same workflow as it is shown in Zemp et. al 2013:*

1) *Deriving the volume change (see our equation 1) is equally performed in Zemp et al 2013 using the maximum extent of the glacier:*

$$\Delta V = r^2 \sum\nolimits_{k=1}^{K} \Delta h_k, \tag{4}$$

where $K$ is the number of pixels covering the glacier at the maximum extent, $\Delta h_k$ is the elevation difference of the two grids at pixel $k$, and $r$ is the pixel size. Geodetic surveys are ideally carried out at the end of the ablation season, si-multaneously with the glaciological survey, and preferably

2) *We use the average glacier area during volume-to-mass-conversion (see our equation 2) as it was proposed in Zemp et al. 2013:*

$$B_{\mathrm{geod.PoR}} = \frac{\Delta V}{\overline{S}} \cdot \frac{\overline{\rho}}{\rho_{\mathrm{water}}}, \qquad (5)$$

where $\overline{\rho}$ is the average density of $\Delta V$, assuming no change in bulk glacier density over the balance period, and $\overline{S}$ is the average glacier area of the two surveys at time *t0* and *t1* assuming a linear change through time as

$$\overline{S} = \frac{S_{t0} + S_{t1}}{2}. \qquad (6)$$

**Line 213. The glaciological data is mentioned in 3.2. you could define ELA in chpter 3.2. on line 144 and write how it is calculated from the equilibrium line. And remove 'For the Vernagtferner, the altitude of the equilibrium line altitude (ELA) is known from intense glaciologic surveys on an annual basis (BAdW, 2019).'**
*L149: Thank you for your advice. We moved the explanation of the ELA data to chapter 3.2.The sentence For the Vernagtferner… was removed.*

**Line 214, you mean the mean ELA for these three periods? Rewrite: 'The observed ELA of Vernagtferner varied between xxxx and yyyy in the study period, the mean ELA was … for the period ….**
*L197: Thank you for your suggestion. We do not think that giving mean and range information makes much sense in this case since we only have two periods and one mean value for the overall period. However, we added 'averaged over the entire period' so that the reader clearly understands that the ELA for the period 2009-2018 is the mean ELA of the respective period. New sentence:*

*"In contrast to most of the glaciers within our study area, the ELA (Sect. 3.2.) of the Vernagtferner is known and lies at 3217 m.a.s.l. for the period 2009-2015, 3278 m.a.s.l. for the period 2015–2018 and, averaged over the entire study period, at 3237 m.a.s.l. for 2009-2018."*

**The variation from 3217 to 3278 to 3237 is very small so how much will the calculation impact the result? Could comment in the text.**
*L197: Thank you for this comment. Please note, that because of the area-height-distribution of the Vernagtferner, those small changes of the ELA have a large influence on the glacier mass balance and on the AAR. To underline this, we derived a rough estimate of the "AAR". We derived the area above the respective ELAs and divided those areas through the total area of the Vernagtferner:*

| ELA [m.a.s.l.] | AAR [%] |
|---|---|
| 3217 | 29.3 |
| 3237 | 24.8 |
| 3278 | 14.8 |

*For other glaciers that are thinner and longer, a change of 60 m of the ELA will not make a big difference, but for the Vernagtferner this will change a lot. In our opinion, this also underlines the great advantage of integrating the ELA for the volume-to-mass conversion (thus the density assumption).*

*We added this information to the discussion:*

*" For the Vernagtferner, ELA increased 61 m between our two study periods 2009 – 2015 and 2015 – 2018. This relatively small change in elevation changed the AAR of this glacier by about 15% and thus has an influence on the bulk density that should not be neglected."*

**Line 224. The data should be homogenised, not only an error estimation. A bit confusing writing.**
*Thank you for your comment. We hope the new version of this paragraph is clearer:*

*L205: "To allow a comparison of the geodetic and glaciologic mass balances, both datasets were reanalyzed independently according to the Steps 1 to 4 in Zemp et al. (2013): Datasets were homogenized (Sect. 3.2 and 4.1.1.), annual glaciological mass balances were accumulated to the periods 09/2009-09/2015, 09/2015-09/2018, and 09/2009-09/2018, mean annual mass balances of the respective periods (Sect. 3.2. and 4.1.2) as well as systematic and random errors for all geodetic datasets were derived (Nuth and Kääb, 2011)."*

**Line 229. Calibration is done when needed so you should comment in the result if this is needed – after comparing the results. so state her that you homogenised the data and quantified the errors.**
*L209. As we argue in the ms, systematic errors are of great interest to our paper so calibration of those would not make sense. As a consequence, we think that the following sentence, that is now in our manuscript, is important to clarify to the reader why no full reanalysis was performed.*

*"Because one main objective of this paper was to analyze systematic differences between the two methods, iterative adjustment and calibration of the data (Step 5-6, Zemp et al. (2013)) was not performed."*

**Line 259. This method -> be specific, write DDF method if this is what you mean. Details on your work should be in the paper.**
*L228:. Changed from "this method" to "DDF method". Details on our work can be found in our Paper, Sect 4.2.*

**Line 262. Do you really calculate the geodetic mass balance for the correction periods, do you now calculate a correction? In table 2 you refer to correction parameters, I would liked to see the result.**
*We calculate the geodetic mass balance for the correction periods and add/substract those from the original geodetic mass balance as shown in table 2.*
   1) *We compute the surface elevation change between our UAV-survey and the aerial survey 2018 for all altitudinal bands and multiply those values with our density assumption for each altitudinal bands. The resulting correction function ($B_{geod,e,t=corr}$) can be interpreted as the geodetic mass balance of this month in 2018 (Figure 3).*

2) *By dividing this $B_{geod,e,t=corr}$ through the positive degree day sum (again for each elevation change this value changes due to the vertical lapse rate) and the number of days we derive our $DDF_e$. This degree day function [m w.e. a-1/(°C\*d)] can be interpreted as the (geodetic) mass balance change that occurs per day per positive degree for each individual elevation band. Please note, that this function is only valid in the ablation period of the hydrologic year. Since all our correction periods are between August and September, this is valid. (Equation 4)*

3) *By multiplying this $DDF_e$ – function with the individual (altitude-dependent) positive degree day sums and the number of days of the respective correction period, we derive the geodetic mass balance of each individual correction period (Equation 5 and plot below).*

[Figure]

4) *Those individual correction functions are then added/subtracted from the original geodetic mass balance of the entire study period. (See table 2)*

*E.g.* $B_{geod,e,09-18} = B_{geod,e,\ 09/2009-09/2018} - B_{geod,e,t=1} + B_{geod,e,t=3}$

*As you can see in the plot above, the corrections for the 27 days in 2015 will have the greatest influence on the geodetic mass balances. This is because this correction period is the longest and has the highest PDD.*

*For the geodetic mass balance 09/2009 – 09/2018, where the two correction functions of 2009 and 2018 of the plot above are subtracted from each other, the correction will only slightly change the geodetic mass balances.*

*Our comparison with the glaciologic data (Figure 8) shows that those corrections result in geodetic mass balances approaching to the glaciologic mass balances. This is the case for all periods, even though meteorologic conditions have been different between the three years.*

**Fig 4. Why not have 2009 and 2018 outlines of all glaciers on this figure. The font is not very readable.**

*Thank you for your suggestions. We added the outline of 2009 for the Hintereisferner (and the respective survey date). Font size was increased. We do not want to add further outlines of other glaciers because we want the reader also to focus on the gradient of surface elevation change around the glacier tongues. If we would add further outlines, they would partly overlay those gradients.*

**Fig. 5. The font not very readable. Which glacier tongue in the center-right, add name of ID.**

*Changed to "second glacier tongue from the right" and added a white dotted circle to clarify the position of the glacier tongue of interest. In our opinion, the font should not be bigger, and its readability is good. Please note, that all glacier tongues, that are visualized with outlines belong to the same glacier, the Hochjochferner. Thus, they all have the same ID even though they are no longer spatially connected (because of glacier retreat).*

**Fig. 6. This figure text is very short.**

*Instead of 'topography' we now enumerate the different plots b-d.*

**Fig. 9. Is this the corrected values?**

*Yes, see Sect 4.2., where the variation function was described (Equation 7). The variation function is the difference between the corrected geodetic and the glaciologic MB. Clarified in the caption.*

**Fig. 10. What is the scale used in this figure, it is difficult to see the different categories. Is it uneven colour scale such as rainbow map (https://www.nature.com/articles/s41467-020-19160-7)? It could be better to have the scale on the figure or use discrete classes, e.g. 8 or 10 classes. The table could be taken out and showed in a table.**

*Thank you for this valuable comment on Figure 10. We tried a visualisation using discrete classes (Range per class: 0.3 m w.e. a-1; see below). However, we feel like this visualisation would reduce the amount of information that the reader could get out of this figure. And, more importantly, the gradient of smaller features, for instance of debris-covered areas are important to see, since they are part of our discussion.*

*Concerning the choice of colours-scale. A suitable colour-scale must have three colours in order to be able to distinguish between the three important cases $B_{geod} > B_{glac}$, $B_{geod} < B_{glac}$ and $B_{geod} = B_{glac}$. Thus, any other choice of colour would not improve the scale. In our opinion, a linear scale, such as the colour-scale chosen, is appropriate.*

*To increase understandability, we removed "Low" and "High" from the legend and added the "0-value" to underline the linearity of the colour scale.*

[Figure]

*We agree that the table is better seperated from the figure and applied the changes in the manuscript.*

**Line 454. Here you talk on height changes but the figure shows changes in m w.e.**

*Thank you for finding this mistake. Corrected accordingly.*

**Line 461. 'All glaciers' are written two times in this sentence, remove one occurrence.**
*Again, thank you for finding this typo. The diligence with which you read the document is even more impressive considering that you only had the authors tracked changes version available. Thanks you very much for that.*

**Line 518. It is not easy to see the dead ice body from figure 4, even if it is marked. I suggest to show it in a subset figure, e.g. add a frame so it is possible to see it.**
*We are not able to extract the exact outline of the dead-ice body only with the data we have. Thus, we would like to remain with only referring to its existence and the surface elevation changes that obviously occurred because of the existing dead ice body. We checked the manuscript for misleading statements concerning this dead-ice body and removed the square from figure 4.*

**Line 538. It is the resulting geodetic mb that must be considered with caution.**
*L439: This was badly written. Thank you for pointing this out. New version of this sentence: Thus, the density assumption and the derived geodetic mass balances for this period must be considered with caution.*

**Line 540. Why do you not use it for the other glaciers then, or try to estimate it with the data you have on ELA and orthophotos or retrieve ELA using satellite images?**
*Within this paper we developed and tested a robust method to correct geodetic mass balances temporally. We compared our results to glaciologic data spatially and quantitively and derived (uncorrected) geodetic mass balances of 23 different glaciers. We showed that there is great potential if all existing photogrammetric data sets were analyzed accordingly. In our opinion, applying our method on other glaciers, including determining the altitude of the ELA from satellite imagery etc and collecting meteorological data for the respective glaciers would be beyond the scope of this ms.*

**Line 636. How does your method compare to other methods such as mass balance modelling being used to correct for acquisition dates?**
*We contextualized our correction method as well as our density-function and thus added a comparison with other studies in the introduction (see below) and discussion:*

*Density function:*
*"The presented altitude-related density function is easily applicable and need low computational effort and therefore greatly complements other existing density conversion factors, that account for e.g. firn compaction processes or rely on classification methods or modelling (Pelto et al., 2019; Reeh, 2008)."*

*Correction function:*
*"Such corrections can for instance be applied by using a simple degree-day model (Belart et al., 2019) or field measurements (Fischer et al., 2011). These methods, however, are either not suitable for retrospective corrections where no field data was collected or do not account for the spatially distributed, glacier specific accumulation and ablation patterns of each glacier (Huss et al., 2009)."*

**Line 645. What about Lidar surveys? And aerial imagery with poorer contrast (e.g. ice caps)?**

*This was a very important Feedback in our opinion and that we incorporated within our discussion. The presented method to adjust geodetic mass balances is not limited to UAV and/or photogrammetry. It can be used with many other geodetic survey setup. We specified this in our discussion:*

*L473: "The required geodetic survey is neither limited to a platform, nor to a sensor. Thus, all geodetic survey configurations that allow the determination of geodetic glacier mass balances from DSM differences are suited for deriving the presented correction function. We used a dedicated photogrammetry-based UAV survey, flexible and low-cost, that enabled a correction with high spatial resolution. We used a dedicated photogrammetry-based UAV survey, flexible and low-cost, that enabled a correction with high spatial resolution."*

**Line 648. But this was only done for one glacier. Using UAV in addition to aerial imagery for all these glaciers is not manageable. And you need to show how you can use information from one glacier to the others to make this argument valid.**
*See L467, where we explain how our method can be transferred to other glaciers:*

*"For transferability to other glaciers, individual correction functions must be determined for each glacier individually, as these are directly related to glacier-specific slope gradients, orientation, the height of the glacier tongue, and area-height distributions (Fig. 11). To determine such individual correction functions, temperature information (for the correction period as well as all periods to which the correction is applied) and an additional geodetic survey is needed to estimate the surface elevation changes during the correction period. The length of the correction period was one month within this study, however we do not expect poorer results if varying this period by 1-2 weeks"*

*Please also note, that with the increasing range and flight time of UAV, especially fixed-wing UAV are suitable to survey large areas and thus many glaciers could be corrected accordingly and that as soon as one correction function is derived, our method allows the retrospective correction of multiple time periods for each glacier.*

**Appendix or supplementary material in the end of ms. As it is only one table you could have it in the manuscript itself as ordinary table.**
*Applied. Please note that we also doublechecked Figure 11 as well as the table 4 for equivalent writing of all glacier names.*

**Table text above the table. Are this corrected values? You need to add more info.**
*L395, added: Geodetic mass balances were derived by using a fixed density factor.*

**Use dot (.), not comma (,) as decimal operator throughout in paper**
*Checked the manuscript again.*

---

## Author Response (AR3)

Dear Etienne,

Thank you for your answer and your comments on our revised manuscript.

You can find our detailed point by point answer below. We also uploaded a tracked-changed version of our manuscript.

Our main changes focus on:

- Missing temporal correction has high influence on geodetic mass balance

- Error contribution of horizontal shift of DOM 2009

- Discussing that presented methods need further testing on new datasets

Please do not hesitate if you have any further questions.

Looking forward to your final decision.

Best regards,

Joschka Geissler and Co-Authors
* * *
L11: this is not useful for the TC reader. Funders/Projects should only be named in the acknowledgment section.

Following your comment, we only name the project in the acknowledgments.

L 21: debris cover

We defined the following spelling convention and double-checked consistent writing:

…debris cover…
…debris-covered areas…

L 25: this not only retreat but also mass loss. Maybe a more generic term? wastage?

We prefer remaining consistent with the manuscript, in which we refer to glacier retreat and mass loss. We thus added "and mass loss" and did not use a new, generic term.

L66: If you read the conclusion of Huss, 2013 (not only the abstract) you will see that 5 years is recommended in fact. Maybe "three to five" years?

Thank you for pointing this out. We have incorporated your comment. L452 was adapted accordingly to be consistent.

L158: As TOM is not a usual acronym in glaciology I really recommend to remove it from the entire text.

We now use the more generic term orthophoto. Checked the manuscript and figures for consistency.

Figure 3: Authors should provide the exact dates of survey in the legend of this figure. This is key for the easy understanding of the figure.

We now provide the exact dates of survey in the legend and the caption. Changed L215 for consistency.

Table 2: Unclear to me why this is late August here when it is 30 sept for all other years (2009 and 2018). Can authors clarify this? The title of this column should be improved it does not correspond to "Acquisition date photogrammetric data". 30 sept is the end of the glaciological year.

Thank you for your valuable comment. Here, you found two mistakes that were corrected as follows:

- Header was renamed from "Acquisition date photo. Data" (wrong!!) to "Correction periods"
- Row 2 must be 30.09. 2015 (typo!). Also changed the corresponding number of days. We double-checked our calculations (used the right period here).
- We changed some explanations to clarify the method.

Figure 4: change to 2500 m (no dot)

We changed this. Thank you for finding this typo.

L 258: clarify that it was not corrected (right?)

Even though we were unsure what you mean by "corrected", we did take into account this comment. A) We clarified that error assessment was conducted using uncorrected DSM differences and B) extended our discussion regarding the remaining error depending to the aspect. L436pp (See also comment below)

**Figure 6: This indicates that for North (or nothwest) oriented glacier, the error would be almost 1 m so 0.15 m/yr given the 6 years time interval. Is this fully included in the error bar? Did the authors verify the same plot for the other periods of interest. If similar biases are found for the 2015-2018 period, the error could be almost 0.3 m/yr for the annual rate of change. Far from negligible.**

Thank you for your feedback on our error assessment. We discussed your feedback and decided to add the results of the period 08/2015-09/2018 to the figure 6 for the sake of completeness although errors are much smaller for this period and no relation to the aspect can be found. We revised some sentences in the methods and results and, more importantly, improved our discussion.

In general, the accuracy assessment conducted (following Rolstadt 2009) relies on the determination of range and sill – values within the semivariograms (see sect. 4.3 for more information on this method and the assumptions made). Since semivariograms have a sill for all our datasets, we assumed that the prerequisites for this method were fulfilled.

However, you are right that for a north-facing (or northwest-facing) glacier the error related to the aspect can get quite high:
Following your example, for the period 09/2009-08/2015, the error of the DSM difference would – for a N-facing glacier – be **0.13** m ice $a^{-1}$. Given the mean surface change of all glaciers within that period (-0.5 m ice $a^{-1}$) this would be 26%. We agree, that this error is not negligible.

For the same period, for a (hypothetic) north-facing glacier of 2.6 km² surface area (mean surface area of all glaciers), the confidence interval would lie at **0.1 m** ice $a^{-1}$. Thus, the possible error would be larger than our error bars due to the relation of the error to aspect. This fact was already discussed in earlier versions of our manuscripts (L430 in the last version published) and is now discussed with even more detail within our discussion (L436pp). This problem could have been addressed with a complete coregistration of all DSMs.
The error was not fully captured by our methodology possibly due to inaccurate assumptions. However, it does not affect the methodologies presented in the manuscript. For the Vernagtferner (on which the focus of our work is), having orientations between E and SW, this error is negligible.

We have taken special care to ensure that the reader is always able to assess the errors involved. Additionally, we checked all geodetic glacier mass balances and other numeric results presented in the manuscript, if the confidence interval covers the potential errors caused by the relation to the aspect. We highlighted those glaciers in table 4, where our confidence intervals underestimate potential errors due to the relation to the aspect.
For your information, for the period 09/2009 – 09/2018, the horizontal shift of the DSM 2009 also results in an error related to the aspect. However, these errors are smaller (max median vertical error 0.098 m $a^{-1}$) and all errors for this period lie within our error bars.

[Figure]

*Figure 1: Relation of the vertical error to the aspect for the period 09/2009-09/2018*

Figure 7: Authors should change the X-axis so that all values are shown (some curves are cut)
We changed the figure accordingly.

L345: This seems a very large difference to me (but not impossible). Can authors double check?
There was an error involved in the computation of the debris cover and dead ice body volume change that was corrected (L345+L432) Thank you for your attentive reading.

Table 3: Title of the table need improvement
We improved the title of table 3.

L380: and also the quadrupling is partly due to the lack of sesonal corrections. This must be noted.
L 390: The illustration of my statement above. The quadruple is now a doubling "only" (in agreement with the glaciological data). This really needs to be discussed as some readers may jump to an erroneous conclusion. This is also the issue when comparing too short time period, like only 3 years here. Authors need to tell that the magnitude of the acceleration in mass loss is not very well constrained due to this.
We agree, that the quadrupling is influenced by the missing temporal correction. This fact has already been noted in the latest version of our manuscript (L381 L397 of the last manuscript):

*It must be noted that there was no correction applied for the acquisition times. Accordingly, the mass balances do not refer to glaciological data. (…) If a time correction is assumed to have a similar influence on the annual mass balances of other glaciers as it had for the Vernagtferner (…)*

We have addressed your comments as follows:

- We have merged Sect. 5.3 so that both of our statements above will be read by the reader without an interruption of a figure and a table.
- We have included this issue into the discussion L469
- Improved this section in the manuscript L390pp

Please note, that we improved the terminology of the geodetic mass balances and ensured consistency within this revision:  We always refer to temporally uncorrected geodetic mass balances with year and month, e.g. 09/2009-08/2015. This is also the case for the legend in Figure 11 where we show the months of the underlying

time periods. For corrected periods, we only provide the years (e.g. 2015-2018). We now refer to "temporally corrected geodetic mass balances" as "annual geodetic mass balances" since they were recalculated to an annual basis and the prefix "annual" is more commonly used within literature. -> L235

Table 4: This table could be moved to an appendix. It is not necessary for the core article. Two decimals seems sufficient given the errors bars.
- This table was removed from the appendix and moved to the core article following the feedback of referee #2 (06.04.2021). We agree with him, that opening an appendix section only for this figure is not needed.
- Changed to two decimals.

L414: What about glaciers in Italy and France? Maybe more diversity in the references?

We added a references that provides a good overview on glacier mass balances of 239 glaciers throughout the alps.

Davaze, L., Rabatel, A., Dufour, A., Hugonnet, R. and Arnaud, Y.: Region-Wide Annual Glacier Surface Mass Balance for the European Alps From 2000 to 2016, Front. Earth Sci., 8, 149, https://doi.org/10.3389/feart.2020.00149, 2020.

authors develop the correction method using a single observation period. So the transferability to another period remain to be tested and fully demonstrated. I would recommend writing the need for further tests in the discussion and proposing to repeat this evaluation of short-term elevation changes in the future.

The fact that the method is robust still need to be demonstrated (e.g., using new surveys as proposed above). Right now its application to new periods or other glaciers is not fully validated.

Thank you for your comment. We use one observation ("correction") period (in 2018) and apply the derived correction to three different periods (September 2009, September 2015, September 2018). Fig. 8 proves that the results are good, also for the retrospective corrections. Thus, in our discussion, we want to underline the possibility of the retrospective correction (with different limitations that are mentioned). However, you are right that further testing of the robustness of this method is needed, using different surveys on different glaciers and periods…

We have made the following changes in this regard:
- We avoid the word robust in this context
- We underline that new surveys are needed to test and evaluate our method. L487

Glaciologic/Glaciological

Following your advice, we now write "glaciological" instead of "glaciologic". Checked manuscript for consistency.